# Amorphous/crystalline interwoven multipods with high Co/Ni activity for wide-temperature-range sodium-sulfur batteries

Tingjiao Xiao[1,2,9], Zhen Fang[3,4,5,9], Nian Ran[6,9], Ronghui Liu[1,2], Yuxuan Gao[7], Jianbo Wu [3,4,5], Jianjun Liu [6], Hua Wang [2], Wen-Feng Lin [8] & Wei Zhou [1,2,8] ✉

Sluggish kinetics caused by 16-electron transfer hinders development of wide-temperature-range sodium-sulfur batteries. Here we report Sn-doped CoNiS multipods with an amorphous-crystalline interwoven structure. Employed as a positive electrode catalyst, the resulting sodium−sulfur battery exhibits a discharge capacity of 1320.8 mAh g$^{-1}$ at 3 A g$^{-1}$ after 1200 cycles at room temperature, together with stable and high-capacity electrochemical performance ranged from −20 to 50 °C. It has been evidenced that the amorphous/crystalline interfaces generated by Sn doping can adjust the microelectronic environment of Co and Ni atoms, optimize their adsorption energy toward sodium polysulfide intermediates through Co−S and Ni−S bonding, and thus decrease the energy barrier of polysulfide conversion. This interfacial regulation efficiently lowers the energy barrier of the rate-determining step and facilitates the overall reaction kinetics over a wide temperature range. This work provides an efficient amorphous/crystalline interface engineering strategy to develop high-performance catalysts.

As an emerging type of low-cost sodium secondary batteries, sodium-sulfur (Na||S) battery uses sulfur as positive electrode and metallic sodium as negative electrode. The sulfur electrode involves a complex 16-electron conversion between chemical energy and electrical energy during charging and discharging processes, which endows the battery high capacity[1,2]. However, the sluggish kinetics associated with the complex multi-electron process and the shuttle effect caused by soluble intermediates of polysulfides cause significant challenges, resulting in low battery capacity, rapid capacity decay, and insufficient energy and power densities[3]. As for the batteries running at room temperature, some sulfur host materials with catalytic effect in sulfur electrodes were prepared, such as 3D carbon matrix, $Ti_3C_2$ quantum dots supported on carbon nanotubes (CNTs), and a few transitional metal sulfides, to obtain considerable specific capacities and cycle life[4–6]. Considering the greenhouse effect and extreme weather we are facing, it's crucial to develop all-climate wide-temperature-range batteries to meet the needs of operation in multiple regions and all seasons. It is worth mentioning that the sodium ion diffusion and the conversion process of sodium polysulfides (NaPSs) can be dramatically reduced at low (sub-zero) temperatures, which leads to polarization and limited discharge capacity. It has been reported that the Na$^+$ diffusion rate drops by ~70 % and the capacity drops by ~30 % when the

[1]School of Chemistry, Beihang University, Beijing, China. [2]Hangzhou International Innovation Institute, Beihang University, Hangzhou, China. [3]Center of Hydrogen Science & State Key Laboratory of Metal Matrix Composites, School of Materials Science and Engineering, Shanghai Jiao Tong University, Shanghai, China. [4]Future Material Innovation Center, Zhangjiang Institute for Advanced Study, Shanghai Jiao Tong University, Shanghai, China. [5]Materials Genome Initiative Center, Shanghai Jiao Tong University, Shanghai, China. [6]State Key Laboratory of High Performance Ceramics, Shanghai Institute of Ceramics, Chinese Academy of Sciences, Shanghai, China. [7]School of Materials Science and Engineering, Beihang University, Beijing, China. [8]Department of Chemical Engineering, Loughborough University, Loughborough, UK. [9]These authors contributed equally: Tingjiao Xiao, Zhen Fang, Nian Ran. ✉e-mail: zhouwei@buaa.edu.cn

operating temperature decreases from 25 to −20 °C[7–9], showing a significant reduction and slowdown of the kinetic rate due to low temperature. Referring to higher temperatures above 40 °C especially at high specific currents, Na||S batteries suffer from accelerated capacity decay and shortened lifespan because of the exacerbated shuttle effect[10,11]. Although the room-temperature positive electrode catalysts in Na||S batteries have made some progress, it remains a great challenge to develop wide-temperature-range catalysts with further enhanced adsorption properties and catalytic activity.

Transition metal sulfides have been proven to be efficient catalysts for metal−sulfur batteries[12]. In particular, the unique electronic structure of cobalt enables a balance between polysulfide adsorption and conversion, ensuring stable catalytic activity over a wide temperature range[13–15]. For example, Dou and his coworkers used cobalt sulfide as an electron reservoir to enhance the activity of the sulfur positive electrode of Li||S batteries, enabling sulfur to be directly reduced to short-chain lithium polysulfides through a simplified redox pathway[16]. Interface engineering can be one feasible strategy to further promote the catalytic and adsorption performances towards polysulfides[17,18]. Among different interface types, the amorphous/crystalline interface could be an ideal structure for Na||S batteries. It combines the advantages of both phases, including the high conductivity and structural stability of crystalline regions as well as the abundant active sites and strong adsorption capacity of amorphous regions[17–21]. For instance, Lee et al. constructed a nanoscale crystalline–amorphous $MoO_3$ heterostructure, which facilitated continuous adsorption and conversion of lithium polysulfides, leading to high sulfur utilization and stable cycling behavior[19]. Moreover, Liu et al. demonstrated that a crystalline–amorphous $Ni_2P/CeO_x$ heterostructure with strong interfacial electronic interaction could accelerate $Li_2S$ deposition/decomposition kinetics and improve long-term cycle life[20]. Considering better electron transfer enabled by the crystalline structure, interwoven amorphous and crystalline structures containing metal sulfides could have great potential for developing suitable catalysts. Traditionally, amorphous structure can be obtained by low-temperature reaction, rapid cooling treatment, and reduction atmosphere growth[4,17,18,22]. Besides addressing the low-temperature kinetics, high-temperature anti-shuttle performance needs to be considered as well. Supported catalysts could be a good choice. On one hand, the substrate with high specific surface area can disperse metal sulfides, serve as a sulfur host, and provide more active sites during the electrocatalytic process. On the other hand, it can afford suitable adsorption energy, suppressing the dissolution and the shuttle effect of the intermediates of NaPSs.

Based on these factors, this work applied Sn doping to interfere with the crystal growth of CoNiS and thus synthesized CoNiS multipod-like structure with amorphous/crystalline interface (A/C-CoNiS) using two-dimensional (2D) MXene sheets as substrate. After loading sulfur, the as-prepared sample was used as both a catalyst and an adsorbent in wide-temperature (all-weather) Na||S batteries. At room temperature, it exhibits a high discharge specific capacity of 1320.8 mAh g$^{-1}$ after 1200 cycles at 3 A g$^{-1}$ with a decay rate of 0.012 % per cycle. At −20 °C, it shows a high discharge capacity of 949.9 mAh g$^{-1}$ after 400 cycles, even at a high specific current of 2 A g$^{-1}$. At 50 °C, it maintains stable cycling performance, with a capacity decay rate of only 0.001 % per cycle over 700 cycles. Control experiments, electrochemical measurements, and density functional theory (DFT) calculations provided solid evidence to confirm the role of amorphous/crystalline interfaces generated by Sn doping in optimising adsorption energy towards intermediates and decreasing the energy barrier of NaPSs conversion, which then boosts kinetics and mitigates the shuttle effect for wide-temperature-range batteries.

## Results and discussion

### Synthesis and characterization of multipod-like A/C-CoNiS catalyst

Figure 1a illustrates a typical synthesis route for preparation of A/C-CoNiS grown on $Ti_3C_2T_x$ MXene. The characterizations of $Ti_3C_2T_x$ MXene were shown in Supplementary Fig. 1. Because of having negatively charged surfaces, MXene can easily adsorb $Co^{2+}$, $Ni^{2+}$, and $Sn^{2+}$ ions via electrostatic interaction. By adding hydrazine and heating, $Co_{0.5}Ni_{0.5}Sn(OH)_6$ nanocrystals were then formed on MXene, serving as the precursor inducing the growth of multipod structure (Supplementary Fig. 2). A/C-CoNiS was finally produced through sulfurization process (Supplementary Fig. 3). It's interesting to find that Sn introducing not only induces the formation of unique multipod structure, but also leads to the interwoven crystalline and amorphous structure. Figure 1b shows the high-angle annular dark-field scanning transmission electron microscopy (HAADF-STEM) image of several A/C-CoNiS multipods. Figure 1c shows a uniform distribution of Co, Ni, S, and Sn elements in one typical particle. The Sn content is very low, which was further verified by ICP-AES (molar ratio, Co: Ni: Sn = 1: 0.9: 0.1, Supplementary Table 1). The selected-area electron diffraction (SAED) pattern was selected from the branch tip in Fig. 1c confirms the single-crystal nature of the branch. The atomic image in Fig. 1d reveals the coexistence of amorphous (red frame) and crystalline (blue frame) phases within A/C-CoNiS. The lattice fringes well match the crystal structure of $(CoNi)_9S_8$ with spacings of 0.22 nm corresponding to the (331) planes of $(CoNi)_9S_8$[23]. To better understand the structure, TEM and HRTEM images in Fig. 1e combined with Supplementary Fig. 4 disclose that the branches of the particle are assembled by many directional-growth single-crystal thin rods, in agreement with the SAED result. The right HRTEM image clearly exhibits the amorphous structure and neighboring (311) crystal planes of $(CoNi)_9S_8$[23–25], revealing that A/C-CoNiS particles were composed of some single-crystal branches dispersedly decorated by amorphous grains. Those single-crystal thin rods might provide faster electron transfer during catalytic process[17,26], while the amorphous phase could provide more active sites for NaPSs conversion[25]. Additionally, three other transition metals (Cu, Mn, and Zn) were selected for doping, but no comparable effects were observed (Supplementary Figs. 5–7). To confirm the importance of amorphous/crystalline interfaces, we synthesized crystalline CoNiS (C-CoNiS) for comparison with characterizations in Supplementary Fig. 8. Compared with C-CoNiS, the ratio of $Co^{2+}/Co^{3+}$ increases from 1.5: 1 to 1.9: 1 while $Ni^{2+}/Ni^{3+}$ increases from 0.7: 1 to 1.2: 1 (Fig. 1f and Supplementary Fig. 9). The reduced chemical valence state of Co and Ni demonstrates the effect of Sn doping on adjusting the electronic microenvironment of Ni and Co atoms. Figure 1g shows that A/C-CoNiS has a stronger electron paramagnetic resonance (EPR) signal at g = 2.003, indicating the sample has more sulfur vacancies[27]. That also explains why Ni and Co of A/C-CoNiS show decreased valence states to meet the charge balance. That is, we generated more sulfur vacancies by Sn doping because of partial amorphization of CoNiS. In addition, nitrogen adsorption−desorption analysis confirms that A/C-CoNiS possesses a larger surface area and a mesoporous structure (~3.9 nm), which facilitates electrolyte infiltration and ion diffusion (Supplementary Fig. 10).

### Na||S battery performances operated from − 20 to 50 °C

The coin cells with A/C-CoNiS/S and C-CoNiS/S electrodes were tested within a voltage window of 0.2 - 2.8 V. Supplementary Fig. 11 shows that A/C-CoNiS/S achieves its optimal electrochemical performance at a sulfur loading of 70 wt %, while the maximum sulfur loading of C-CoNiS/S is 65 wt %. Figure 2a shows the galvanostatic charge-discharge (GCD) profiles of A/C-CoNiS/S and C-CoNiS/S electrodes. The discharge curve of A/C-CoNiS/S exhibits three voltage plateaus at ~2.0, 1.5, and 0.8 V, corresponding to the conversion from $S_8$ to long-chain polysulfides ($Na_2S_x$, 6<x ≤ 8), then to $Na_2S_4$, and finally to $Na_2S$. Upon

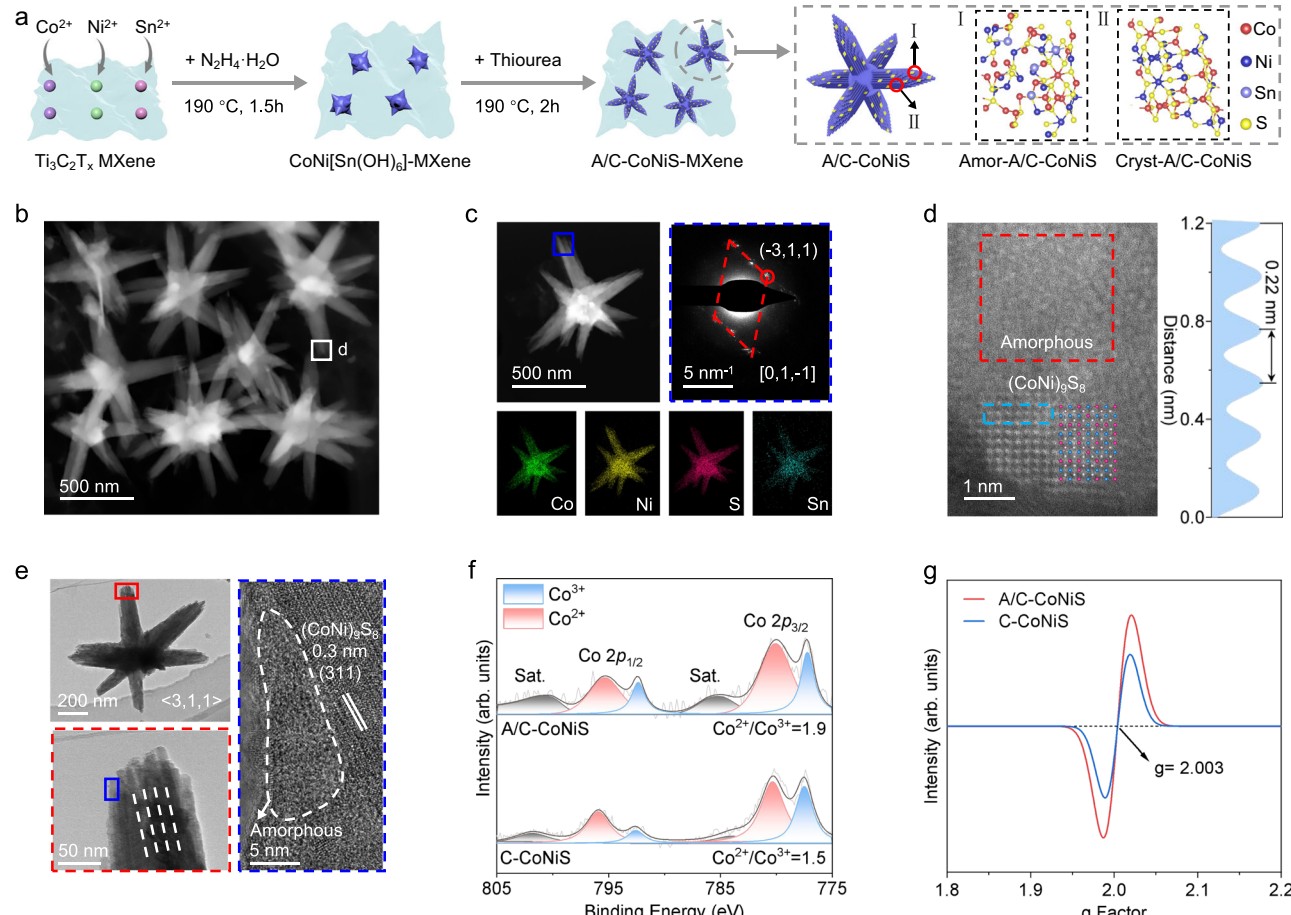

**Fig. 1 | Preparation and characterizations of Sn-doped CoNiS multipods with amorphous/crystalline interface. a** Schematic illustration for the synthesis of A/C-CoNiS grown on Ti₃T₂Tₓ MXene with amorphous/crystalline interfaces. **b** HAADF-STEM image of A/C-CoNiS. **c** One typical multipod with corresponding elemental mapping and SAED pattern of the tip under TEM mode. **d** Magnified HAADF-STEM image of the selected region in Fig. b and intensity profile corresponding to the blue frame. **e** TEM and HRTEM images of A/C-CoNiS. **f** High-resolution Co 2*p* XPS spectra of A/C-CoNiS and C-CoNiS. **g** EPR spectra of A/C-CoNiS and C-CoNiS.

charging, two main plateaus appear at 1.7 and 2.2 V, corresponding to the stepwise oxidation of Na₂S to Na₂S₄ and then to Na₂Sₓ[28]. As marked, the voltage difference (ΔE) between charge and discharge platforms is 0.92 V for A/C-CoNiS/S, much lower than 1.28 V of C-CoNiS/S, indicating better reversibility[29]. Compared with C-CoNiS/S, both longer discharge plateaus and decreased ΔE of A/C-CoNiS/S also reveal a faster kinetic rate. A/C-CoNiS/S exhibits a discharge capacity of 1444.4 mAh g⁻¹ and an initial coulombic efficiency of 98.9 % at the specific current of 1 A g⁻¹, higher than that of C-CoNiS/S (799.9 mAh g⁻¹). It indicates that the amorphous/crystalline structure provides a better catalytic effect and thus effectively enhances the utilization rate of sulfur. In Fig. 2b, A/C-CoNiS/S shows a high capacity of 1437.6 mA h g⁻¹ after 250 cycles at 1 A g⁻¹ with a capacity decay of 0.002 % per cycle and a nearly 100 % coulombic efficiency. By contrast, C-CoNiS/S only achieves a discharge capacity of 730.4 mAh g⁻¹ with a capacity decay of 0.035 % per cycle. Supplementary Fig. 12a shows that the sodium-ion storage performance of A/C-CoNiS contributes minimally to the overall capacity, indirectly indicating that the main contribution comes from the redox process of sulfur. The MXene/S and Ketjen Black/S electrodes deliver capacities far below A/C-CoNiS/S, highlighting the catalytic advantage of A/C-CoNiS (Supplementary Fig. 12b, c). Even though we increased the sulfur loading to 3 mg cm⁻², A/C-CoNiS/S can still give a capacity of 523.2 mAh g⁻¹ after 400 cycles at 1 A g⁻¹ and a nearly 100 % coulombic efficiency in Fig. 2c, comparable to previously reported values (306 ~ 373 mAh g⁻¹)[17,30,31]. Only one close

capacity (529.6 mAh g⁻¹) can be found with sulfur loading of 3 mg cm⁻², but obtained at a lower specific current (0.84 A g⁻¹)[32]. The A/C-CoNiS/S electrode maintains a discharge capacity of 1146.9 mAh g⁻¹ after 50 cycles at 0.2 A g⁻¹ with an E/S ratio of about 10 μL mg⁻¹ (Supplementary Fig. 13). Figure 2d–f show favorable rate performances of A/C-CoNiS/S. All GCD profiles display similar and distinct plateaus without significant capacity decay in Fig. 2d, indicating favorable rate performance and rapid kinetic rate of NaPSs conversion. C-CoNiS/S in Supplementary Fig. 14 exhibits inferior performance with obvious capacity decay even at 5 A g⁻¹. Furthermore, A/C-CoNiS/S in Fig. 2e delivers high specific capacities of 1462.8, 1431.6, 1388.1, 1369.6, 1321.4, and 1276.2 mAh g⁻¹ at 0.5, 1, 2, 3, 4, and 5 A g⁻¹, respectively. Even when the specific current decreases to 0.5 A g⁻¹, the capacity can still be up to 1490.3 mAh g⁻¹, indicating reversible electrochemical behavior and stable cycling. All these values are significantly higher than those of C-CoNiS/S, indicating that amorphous/crystalline interface improves the rate performance by optimizing reaction kinetics. Deduced from the comparison in Fig. 2f, A/C-CoNiS/S shows improved rate performances with high specific capacities, which are competitive with most reported Na||S batteries[3,12,17,28,33–38]. The A/C-CoNiS/S electrode in Fig. 2g delivers a high capacity of 1320.8 mAh g⁻¹ at 3 A g⁻¹ after 1200 cycles with a capacity decay rate of 0.012 % per cycle and a nearly 100 % coulombic efficiency. The above battery data were obtained from three independent parallel tests to ensure reproducibility (Supplementary Fig. 15). The overall battery performance of Fig. 2g was

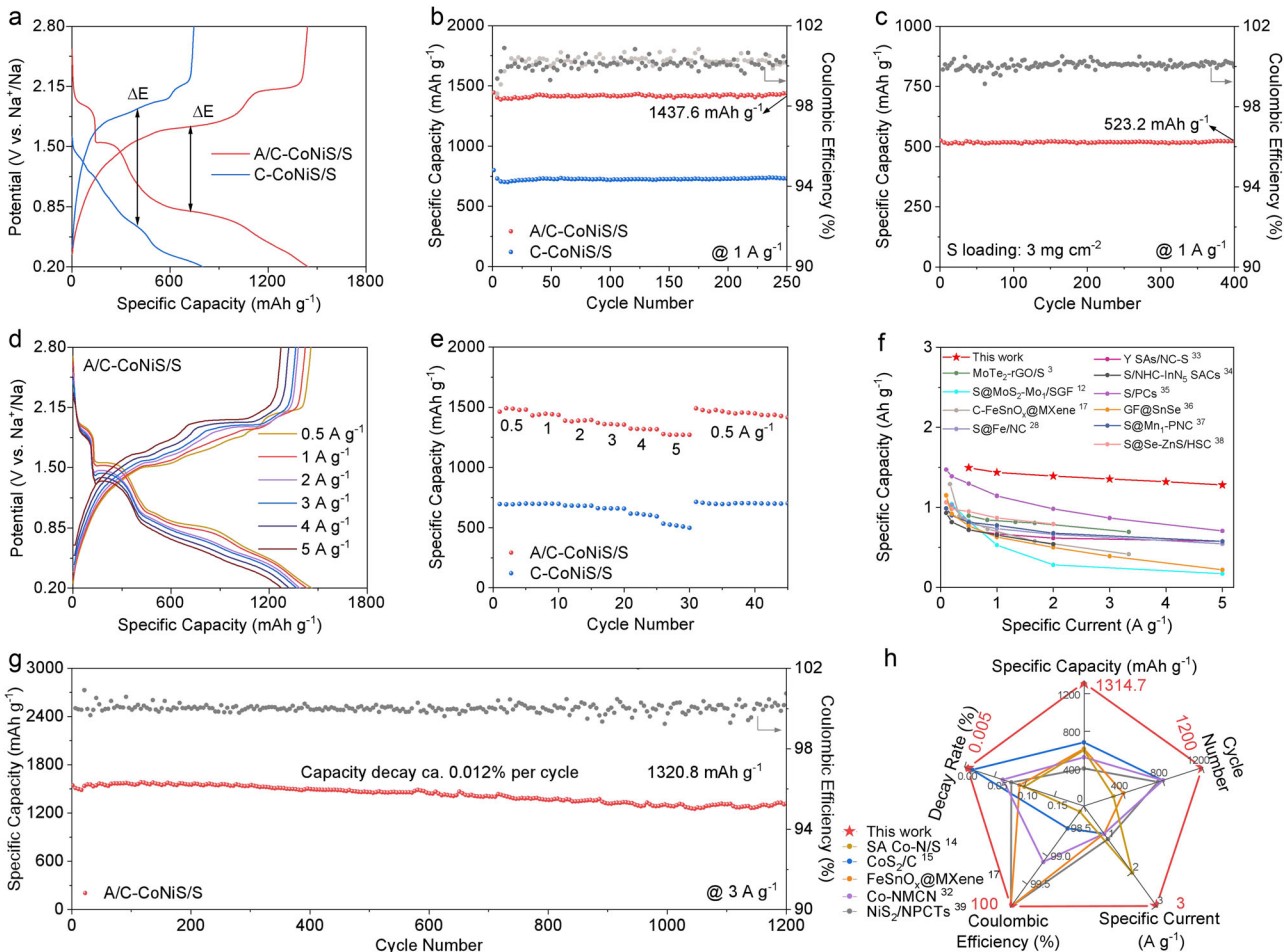

**Fig. 2 | Na‖S battery performance at room temperature. a** GCD profiles and **b** cycling performance at 1 A g⁻¹ using A/C-CoNiS/S and C-CoNiS/S. **c** Cycle stability at a high sulfur loading at 1 A g⁻¹. **d** GCD profiles and **e** rate performance of the two samples at specific currents ranging from 0.5 to 5 A g⁻¹. **f** Comparison of the rate performance of our work with previously reported sulfur electrodes. **g** Long-term cycling stability of the A/C-CoNiS/S electrode at 3 A g⁻¹ over 1200 cycles. **h** Comprehensive evaluation on the overall battery performance using A/C-CoNiS compared with reported Li‖S and Na‖S batteries using other materials (Supplementary Table 2).

improved compared to the counterparts in Fig. 2h, covering specific capacity at certain cycle rate after cycling, cycle number, cycle rate, capacity retention per cycle, and coulombic efficiency[14,15,17,32,39]. Many previously reported batteries exhibit final capacities below 700 mAh g⁻¹ at specific currents lower than 3 A g⁻¹[15,17,32,39–41]. More details can be found in Supplementary Table 2. The radar plot highlights the competitive specific capacity and rate capability of our sample in comparison with reported batteries.

Given the extreme temperatures in real environments, the electrochemical performance tests were conducted at −20 °C and 50 °C to further confirm the boosted reaction kinetics and the mitigated shuttle effect of A/C-CoNiS. First, we carried out low-temperature tests to check the kinetics. Figure 3a shows the change of cyclic voltammetry (CV) peaks between room temperature (25 °C) and low temperature (−20 °C). As for the selected oxidation peak of O₂ (Na₂S → Na₂S₄), it's clear that decreased temperature only makes A/C-CoNiS/S slightly shift by 0.09 V while making C-CoNiS/S shift more positively by 0.20 V, indicating A/C-CoNiS provides a much easier conversion process for NaPSs at low temperatures. The CV curves at 25 °C and −20 °C were shown in Fig. 4a and Supplementary Fig. 16. Figure 3b shows the rate performance of A/C-CoNiS/S at −20 °C. The A/C-CoNiS/S electrode delivers high specific capacities of 1081.4, 1027.6, 936.6, 889.1, 861.0, and 828.6 mAh g⁻¹ at specific currents of 0.5, 1, 2, 3, 4, and 5 A g⁻¹, respectively. When the specific current restores to 0.5 A g⁻¹, the

discharge capacity can still be up to 1051.1 mAh g⁻¹ with almost no capacity loss. For comparison, the capacity of C-CoNiS/S decreases dramatically at 2 A g⁻¹, showing no future application in low-temperature batteries. The comparison of GCD curves of A/C-CoNiS/S and C-CoNiS/S electrodes in Supplementary Fig. 17 also confirms the reliable low-temperature performance of A/C-CoNiS. Besides, A/C-CoNiS/S shows almost the same GCD curves at 0 °C, −10 °C, −20 °C, and −30 °C in Supplementary Fig. 18a, revealing its practical application covering a wide low-temperature range. A/C-CoNiS/S gives high specific capacities of ~2.5 times for C-CoNiS/S at the same low temperature (Supplementary Fig. 18b), supporting an enhanced reaction kinetics of A/C-CoNiS compared with C-CoNiS at low temperatures. A/C-CoNiS/S provides a high capacity of 949.9 mAh g⁻¹ after 400 cycles at 2 A g⁻¹ at −20 °C in Fig. 3c, with a capacity decay rate of 0.001 % per cycle and coulombic efficiency of 99.6 %. By contrast, C-CoNiS/S delivers 510.1 mAh g⁻¹ after 150 cycles at 1 A g⁻¹ with a decay rate of 0.096 % per cycle. All the above results confirm the competitive low-temperature performance of A/C-CoNiS, including specific capacity, rate performance, and cycling stability.

Then we studied the high-temperature performance of A/C-CoNiS/S. As shown in Fig. 3d, the A/C-CoNiS/S electrode exhibits discharge capacities of 1580.5, 1624.6, and 1652.7 mAh g⁻¹ at 30 °C, 40 °C, and 50 °C at 1 A g⁻¹, respectively. The high capacity can be attributed to

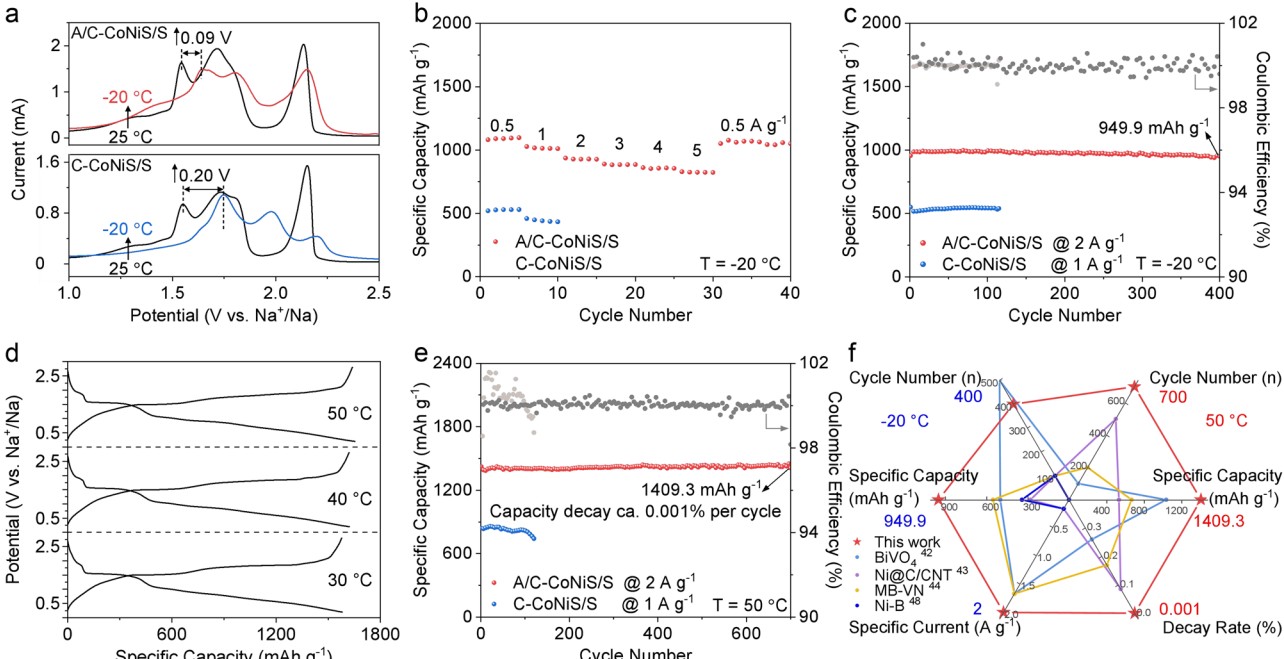

**Fig. 3 | Na∣∣S Battery Performance at low and high temperatures. a** Effect of low temperature on CV curves compared with room temperature for A/C-CoNiS/S and C-CoNiS/S electrodes. **b** Rate performance of A/C-CoNiS/S from 0.2 to 5 A g⁻¹ at −20 °C. **c** Cycling performance for A/C-CoNiS/S and C-CoNiS/S at −20 °C. **d** GCD curves of A/C-CoNiS/S at 30 °C, 40 °C, and 50 °C at 1 A g⁻¹. **e** Cycling performance of A/C-CoNiS/S and C-CoNiS/S at 50 °C. **f** Comparison of overall battery performances at −20 °C and at 50 °C between our sample and previously reported materials (Supplementary Table 3).

the contribution from increased temperature and the catalytic effect. The stability and high capacity of A/C-CoNiS/S were also confirmed over 30–50°C in Supplementary Fig. 19. As is well known, increased electron transfer and ionic migration rates at high temperatures promote polysulfide migration, dissolution, and shuttling, thereby deteriorating electrochemical stability. Thus, we need to check its cycling stability. Figure 3e shows that A/C-CoNiS/S gives an initial specific capacity of 1422.7 mAh g⁻¹ at 2 A g⁻¹, which slightly decreases to 1409.3 mAh g⁻¹ after 700 cycles with an average capacity decay rate of 0.001 % per cycle at 50 °C. For comparison, C-CoNiS/S gives an initial discharge capacity of 839.7 mAh g⁻¹ with an 88.3 % retention rate after 120 cycles. The sudden capacity loss of C-CoNiS/S might be caused by a short circuit influenced by the shuttle effect in the battery at high temperatures. The above battery data were obtained from three independent parallel tests to ensure reproducibility (Supplementary Fig. 20). When compared with reported limited Na∣∣S batteries and even Li∣∣S batteries at a wide-temperature range, our sample of A/C-CoNiS/S in Fig. 3f exhibits a competitive balance between capacity retention and rate capability[11,42–45]. More details can be found in Supplementary Table 3. The overall performance includes low-temperature specific capacity of 949.9 mAh g⁻¹ at 2 A g⁻¹ after 400 cycles, as well as high-temperature specific capacity of 1409.3 mAh g⁻¹ at 2 A g⁻¹ after 700 cycles with a low decay rate of 0.001 % per cycle. All the above experiments and calculations evidently confirm that A/C-CoNiS can afford a faster kinetic rate of polysulfides conversion even at low temperatures and better anti-shuttle effect even at high temperatures. The significant improvement in this material is due to the amorphous/crystalline structure induced by Sn doping.

**Improved catalytic reaction kinetics and adsorption energy**
The faster kinetic rate guarantees higher capacities, better reversibility and rate performance, while the improved adsorption energy can mitigate the shuttle effect and prolong lifespan. Figure 4 then

demonstrates improved kinetic rate and adsorption energy of A/C-CoNiS by electrochemical tests, adsorption experiments, and simulated calculations. The CV curves in Fig. 4a display a series of well-defined reduction and oxidation peaks. Four distinct reduction peaks are observed at 1.97 ($R_0$), 1.53 ($R_1$), 1.17 ($R_2$), and 0.81 V ($R_3$), corresponding to the stepwise conversion of $S_8$ to $Na_2S_x$ ($R_0$), $Na_2S_x$ to $Na_2S_4$ ($R_1$), and $Na_2S_4$ to $Na_2S$ ($R_2$, $R_3$)[16,36]. In addition, two peaks at 0.94 V ($R_{M1}$) and 0.50 V ($R_{M2}$) are ascribed to $Na^+$ insertion into CoNiS[46]. The oxidation peaks at 1.30 ($O_1$, $M_1$), 1.54 ($O_2$), 1.72 ($O_3$, $M_2$), and 2.15 V ($O_4$) correspond to the reverse process accordingly[34]. Deduced from the peak area, the charge capacity of A/C-CoNiS mainly comes from the conversions from $Na_2S_x$ to $Na_2S_4$ and from $Na_2S_2$ to $Na_2S$. Compared to C-CoNiS/S, A/C-CoNiS/S exhibits higher specific current and lower polarization, indicating that the amorphous/crystalline interfaces generated by Sn doping effectively boost conversion kinetics of NaPSs[16,47]. The evolution of the CV curves during the initial cycles is shown in Supplementary Fig. 21. Besides, we also used CV tests on symmetric cells in Fig. 4b and electrochemical impedance spectroscopy (EIS) analyses in Supplementary Fig. 22 to confirm the advantage of A/C-CoNiS in catalytic reaction kinetics. Compared to C-CoNiS, A/C-CoNiS exhibits a narrower voltage gap between oxidation and reduction peaks as well as higher current responses, indicating that A/C-CoNiS has much enhanced kinetic rate of NaPSs conversion[28,48]. In Supplementary Fig. 22, the $R_{ct}$ value of A/C-CoNiS/S is 8.1 Ω, much lower than that of C-CoNiS/S (19.8 Ω), indicating its faster charge transfer[16,49]. In situ EIS tests and corresponding distribution of relaxation times (DRT) analyses were conducted at 25 and −20 °C to evaluate the electrochemical catalytic performance across the full working voltage range (Fig. 4c–f, Supplementary Fig. 23). The time constant (τ) represents the relaxation time of each electrochemical process. As shown in Fig. 4c–f, each contour plot can be divided into four regions corresponding to interphase contact resistance (R1), charge transfer resistance (R2, $R_{ct}$), polysulfide diffusion resistance (R3), and ion diffusion resistance (R4)[34]. At 25 °C, the A/C-CoNiS/S electrode

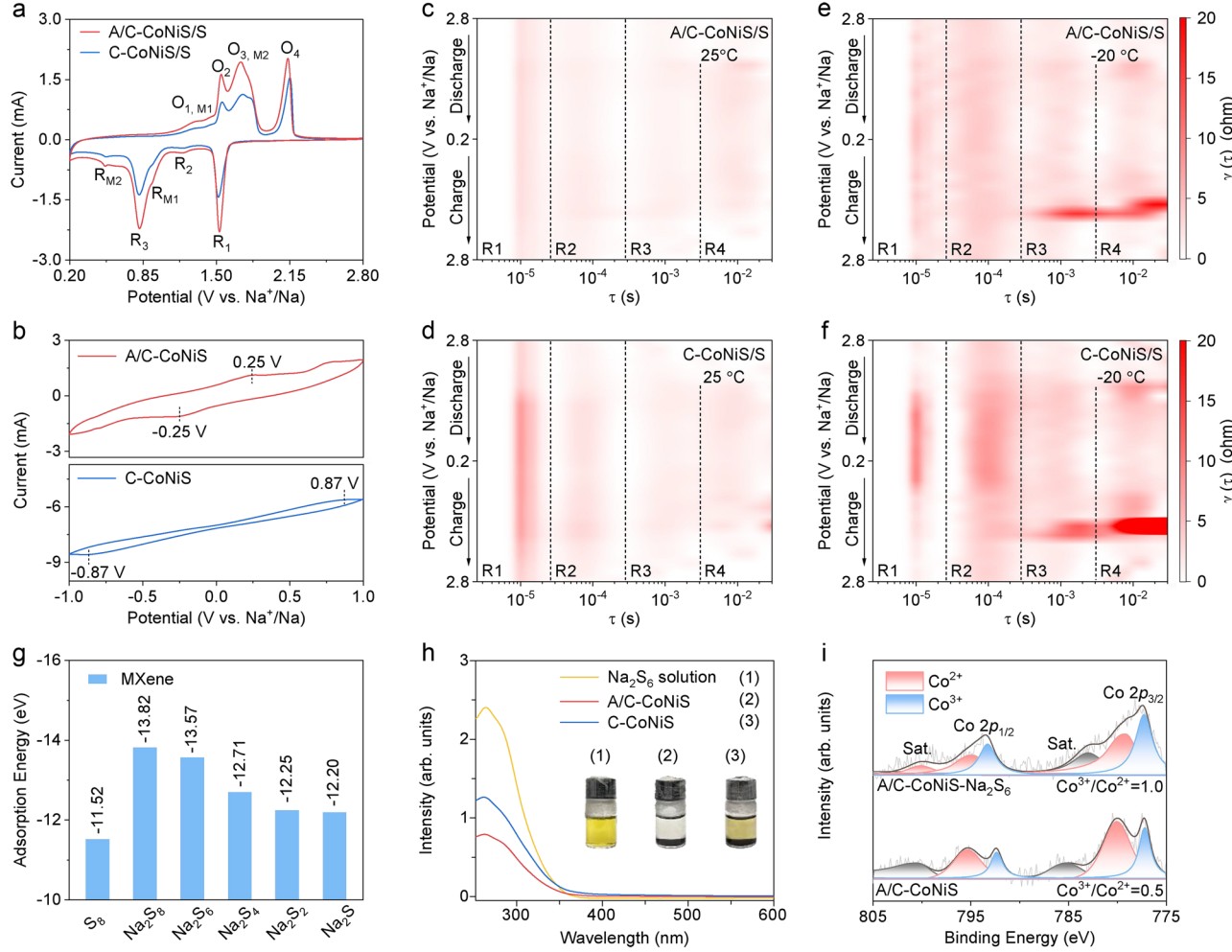

**Fig. 4 | Enhanced kinetic rate and adsorption ability confirmed by experiments.**
**a** CV curves of A/C-CoNiS/S and C-CoNiS/S at 0.2 - 2.8 V vs Na⁺/Na. **b** CV curves of symmetric cells using A/C-CoNiS and C-CoNiS. DRT plots of **c** A/C-CoNiS/S (25 °C), **d** C-CoNiS/S (25 °C), **e** A/C-CoNiS/S ( − 20 °C), and **f** C-CoNiS/S ( − 20 °C) obtained from in situ EIS measurements (voltage interval of 0.2 V). **g** Binding energies of $S_8$ and NaPSs on MXene. **h** UV–vis spectra of $Na_2S_6$ solutions absorbed by A/C-CoNiS and C-CoNiS with optical photos inserted. **i** High-resolution Co 2$p$ XPS spectra of A/C-CoNiS before and after adsorbing $Na_2S_6$.

consistently exhibits lower $R_{ct}$ than C-CoNiS/S (Fig. 4c and d). This trend becomes more pronounced at −20 °C (Fig. 4e and f), indicating the enhanced catalytic activity of A/C-CoNiS/S at low temperatures.

The much-enhanced kinetic process explains high capacity of batteries running at room temperature and low temperatures. Then we carried out adsorption calculations and tests to confirm the anti-shuttle effect and long-term stability at high temperature. DFT calculations in Fig. 4g indicate strong adsorption of MXene towards NaPSs, which might increase the anti-shuttle effect even at high temperatures. Figure 4h shows that A/C-CoNiS exhibits a markedly weaker UV–vis absorption signal than C-CoNiS and the $Na_2S_6$ solution at room temperature, revealing stronger adsorption ability. Quantitative UV–vis adsorption tests (Supplementary Figs. 24 and 25, Table 4) confirm that A/C-CoNiS exhibits greater adsorption capacities especially at high temperatures, indicating an improved anti-shuttle effect. The inserted image shows the obvious color change from yellow to transparent solution by efficient adsorption. The molar ratio of $Co^{3+}/Co^{2+}$ from A/C-CoNiS increases from 0.5 to 1.0 after adsorbing $Na_2S_6$ in Fig. 4i. Similarly, the molar ratio of $Ni^{3+}/Ni^{2+}$ from A/C-CoNiS increases from 0.8 to 1.4 after adsorbing $Na_2S_6$ in Supplementary Fig. 26. The increased oxidation state of Co and Ni elements means the formation of more Co−S and Ni−S bonds with S element from NaPSs. That means the adsorption sites for NaPSs might be Co/Ni atoms. Hence, the as-

prepared A/C-CoNiS with amorphous/crystalline interface can provide optimized adsorption energy, which can effectively alleviate the shuttle effect and guarantee the performance at high temperatures.

Finally, we analyzed the influence of amorphous/crystalline interface in A/C-CoNiS on catalytic reaction kinetics by DFT calculations. As shown in Fig. 5a, the atomic structure covers two parts: the amorphous structure (marked with Amor-A/C-CoNiS) generated by Sn doping, and the side parts with crystalline structure (Cryst-A/C-CoNiS). Figure 5b shows the model of C-CoNiS with only crystalline structure for comparison. Because there are two parts in Fig. 5a, we had to select active sites from both amorphous and crystalline regions. Specifically, we screened multiple possible sites and identified the one with the lowest $S_8$ adsorption energy as the optimal active site (Supplementary Figs. 27–29). As shown in Fig. 5c, d, the rate-determining step from $Na_2S_2$ to $Na_2S$ shows a free energy value of 0.36 eV for C-CoNiS, which decreases to 0.25 eV for Amor-A/C-CoNiS and 0.27 eV for Cryst-A/C-CoNiS. That is, not only the amorphous region (Fig. 5c) but also the nearby crystalline region (Fig. 5d) in A/C-CoNiS exhibit a lower energy barrier for NaPSs conversion. As shown in Fig. 5e, f, the climbing-image nudged elastic band (CI-NEB) calculations reveal that the decomposition barrier of $Na_2S$ is significantly reduced from 0.77 (C-CoNiS) to 0.58 eV (Amor-A/C-CoNiS) and 0.68 eV (Cryst-A/C-CoNiS). The structural evolution of $Na_2S$ decomposition is provided in Supplementary

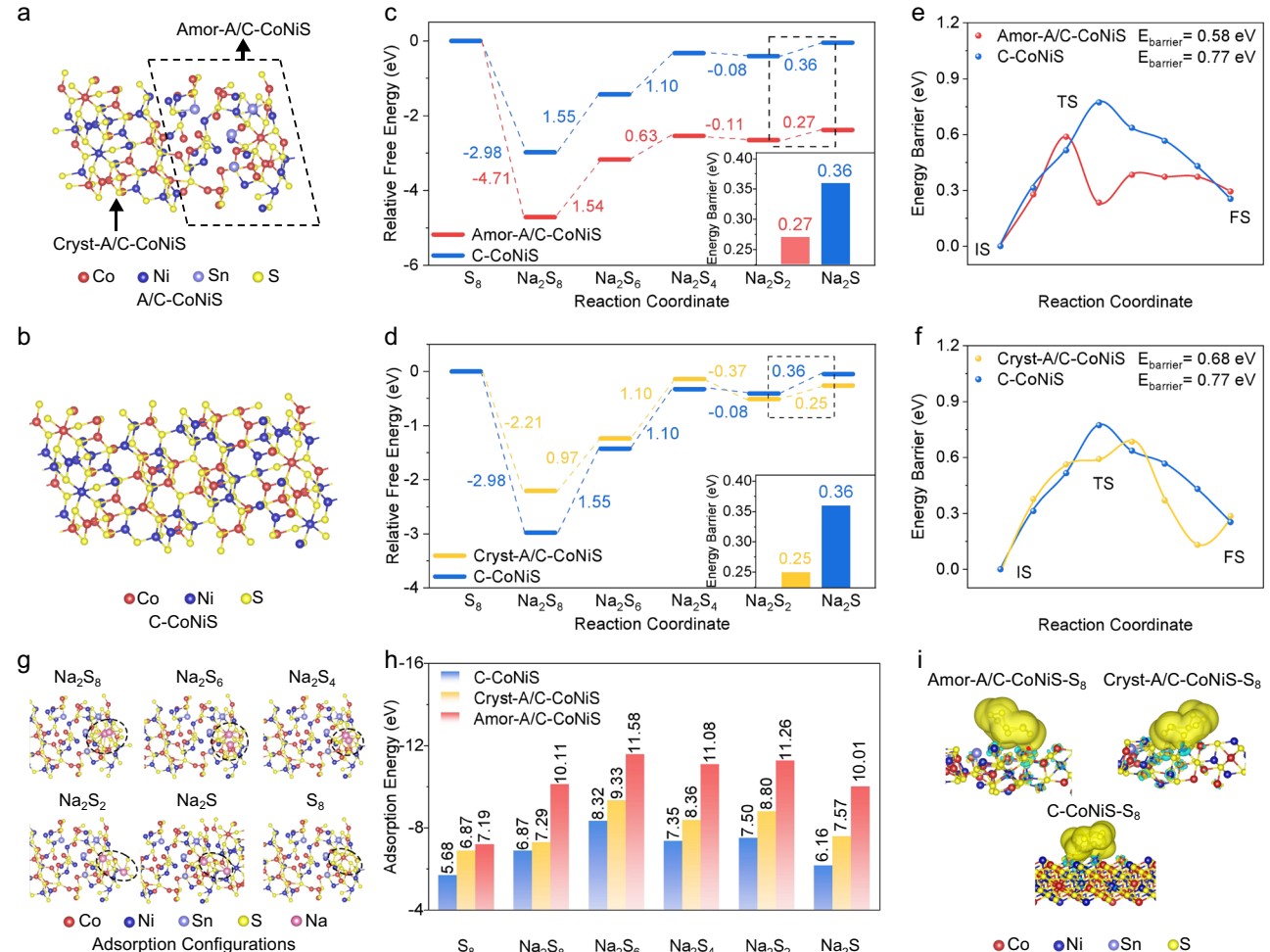

**Fig. 5 | Boosted catalytic kinetics and adsorption ability at the amorphous/crystalline interface revealed by DFT calculations. a** Atomic structure model of A/C-CoNiS. **b** Model of the counterpart C-CoNiS with crystalline structure. Step chart of free energy with active site from **c** amorphous region of A/C-CoNiS (Amor-A/C-CoNiS) and **d** crystalline region of A/C-CoNiS (Cryst-A/C-CoNiS). Both c and d use C-CoNiS for comparison. Energy profiles for Na$_2$S decomposition on **e** Amor-A/ C-CoNiS and **f** Cryst-A/C-CoNiS. Both **e** and **f** use C-CoNiS for comparison. **g** Atomic structure configurations of NaPSs and S$_8$ adsorbed on the surface of A/C-CoNiS. **h** Adsorption energies of S$_8$ and NaPSs on A/C-CoNiS and C-CoNiS. **i** Differential charge density distributions of S$_8$ adsorbed on A/C-CoNiS and C-CoNiS surface with isosurface value of 0.005 e A$^{-3}$.

Fig. 30. Both the amorphous regions and the neighboring crystalline regions of A/C-CoNiS facilitate Na$_2$S decomposition by lowering the energy barrier[50]. These studies verify that the amorphous/crystalline interface generated by Sn doping optimizes both the local electronic microenvironment of metal sites in both amorphous and crystalline regions.

The battery performance, especially high capacity at low temperature and room temperature, could be ascribed to the optimized catalytic reaction kinetics by amorphous/crystalline interface in A/C-CoNiS. Besides kinetics, the performance at room temperature and higher temperatures can also be influenced by the adsorption ability of A/C-CoNiS towards intermediates during the charging/discharging process, which mitigates the shuttle effect and guarantees good running of the battery. The improved adsorption energy of A/C-CoNiS towards NaPSs was then confirmed. Figure 5g displays atomic structure model of A/C-CoNiS adsorbing a series of NaPSs and S$_8$. The atomic structure and adsorption models for Cryst-A/C-CoNiS and C-CoNiS are shown in Supplementary Figs. 31, 32. Figure 5h compares the adsorption energies of A/C-CoNiS and those of C-CoNiS towards S$_8$ and catalytic intermediates. All red and yellow bars are higher than the blue ones, indicating that Amor-A/C-CoNiS and Cryst-A/C-CoNiS

exhibit stronger anchoring and trapping abilities for NaPSs. Figure 5i presents the differential charge density of S$_8$ on different substrates. The stronger charge transfer between A/C-CoNiS and S$_8$ indicates enhanced adsorption capacity[51]. In a word, A/C-CoNiS shows better adsorption capacity in inhibiting the shuttle effect of soluble NaPSs, and thus gives more stable and longer lifespan of all-weather Na||S batteries.

In summary, Sn-doped CoNiS multipods grown on 2D Ti$_3$C$_2$T$_x$ MXene have been successfully synthesized. Partial Sn doping generated amorphous regions in CoNiS, forming interwoven amorphous/crystalline interfaces. After sulfur loading, the A/C-CoNiS/S electrode shows all-weather favorite overall battery performance especially its high capacity, compared to its counterpart CoNiS/S with only crystalline structure. At room temperature, it achieves a discharge capacity of 1320.8 mAh g$^{-1}$ at 3 A g$^{-1}$ after 1200 cycles with a decay rate of 0.012 % per cycle. At −20 °C, it delivers 949.9 mAh g$^{-1}$ at 2 A g$^{-1}$ after 400 cycles. At 50 °C, it maintains a reversible capacity of 1409.3 mAh g$^{-1}$ at 2 A g$^{-1}$ after 700 cycles with a decay rate of only 0.001 % per cycle. Electrochemical tests, adsorption experiments, XPS characterizations, and DFT calculations confirm the capacities and considerable lifespan could be ascribed to the improved catalytic reaction kinetics and

mitigated shuttle effect by improved adsorption ability towards intermediates of sodium polysulfides using the catalyst of A/C-CoNiS. Because the amorphous/crystalline structure can adjust the microelectronic environment of Co and Ni atoms, optimizing adsorption of sodium polysulfides through Co−S and Ni−S bonding, and thus lowering the energy barrier for polysulfide conversion. Moreover, in situ EIS and corresponding DRT analysis verify that A/C-CoNiS effectively accelerates reaction kinetics, particularly under low-temperature conditions. DFT calculation confirms the amorphous/crystalline interface can optimize the rate-determining step of $Na_2S_2$ to $Na_2S$ evidenced by lowering barrier from 0.36 to 0.25 eV (amorphous part in A/C-CoNiS/S) and to 0.27 eV (crystalline part in A/C-CoNiS/S). The amorphous/crystalline interface engineering via element doping provides an efficient strategy for developing suitable catalyst for sulfur electrode in high-capacity and wide-temperature-range Na||S batteries.

## Methods

### Synthesis of A/C-CoNiS and C-CoNiS

The A/C-CoNiS sample was synthesized by a wet chemical method. First, 0.5 mmol $CoCl_2 \cdot 6H_2O$ (Aladdin, 99 %), 0.5 mmol $NiCl_2 \cdot 6H_2O$ (Aladdin, 99 %), 0.05 mmol $SnCl_2$ (Aladdin, 99 %), and 500 mg PVP (Alfa, $M_w$ = 58,000) were dissolved in 100 mL of ethylene glycol (EG, Guangfu, 99.5 %) in a three-neck flask. Subsequently, 10 mg of MXene (JiNan SCMXene Tech. Co., Ltd, 99 %) was added to the solution. After stirring, 2 mL of $N_2H_4 \cdot H_2O$ (Aladdin, 80 %) was introduced. The mixture was heated to 170 °C in an oil bath with vigorous stirring. After 1.5 h, 40 mL of $CH_4N_2S$ (Aladdin, 99 %) solution (4 mmol in EG) was added dropwise, and the flask was kept at that temperature for 2 h. The mixture was then naturally cooled to room temperature, and the resulting black precipitate was washed with ethanol and deionized water by centrifugation (9000 × $g$) several times. Finally, the powder was stored for use after freeze-drying for 24 h. For comparison, the control sample of C-CoNiS was prepared using the same procedure without addition of $SnCl_2$ and MXene.

### Material characterizations

HAADF-STEM imaging was performed using a Cs-corrected FEI Spectra 300 microscope equipped with dual spherical aberration correctors at an accelerating voltage of 300 kV. SEM images were acquired on a FEI Quanta 250FEG field-emission scanning electron microscope operated at an accelerating voltage of 10 kV. XPS measurements were conducted on a Shimadzu/Kratos AXIS SUPRA+ system with monochromatic Al $K_\alpha$ radiation (hv = 1486.6 eV). Pristine samples were prepared and transferred under ambient conditions, whereas polysulfide-adsorbed samples were handled and transferred under an argon atmosphere. Binding energies were calibrated using the C 1$s$ peak at 284.8 eV. XRD patterns were recorded on a Shimadzu XRD-6000 diffractometer with Cu $K_\alpha$ radiation (λ = 1.5418 Å), a step size of 0.02 ° (2θ), and a scan rate of 3 ° min⁻¹. EPR spectra were collected on a Bruker EMXplus-6/1 spectrometer operating at X-band frequencies. UV−vis absorption spectra were obtained using a Shimadzu UV-3600 spectrophotometer over a wavelength range of 200−900 nm. Elemental composition analysis was performed by ICP using a Shimadzu ICPE-9800 system. TGA was carried out on a PerkinElmer STA6000 instrument under a nitrogen atmosphere with a heating rate of 10 °C min⁻¹.

### Preparation of sulfur and metal electrodes

The sulfur (Aladdin, 99.9 %) and A/C-CoNiS powders were mixed at a weight ratio of 7:3 for 10 min using an agate mortar. The mixture was transferred to a stainless-steel high-pressure reactor, sealed under an argon atmosphere in a glovebox, and heated at 155 °C for 12 h to allow sulfur diffusion into the A/C-CoNiS. The working electrodes were prepared by mixing the as-synthesized sulfur composite (active material), CNT (XFNANO, 99.5 %, conductive additive), and poly(acrylic acid) (PAA, Aladdin, 99 %, $M_w$ 45,000, binder) at a weight ratio of 7:2:1 in deionized water (solvent). The solvent-to-solid weight ratio in the slurry was 5:1. The slurry was cast using a doctor blade onto a copper foil current collector (Canrd, 99.8 %, 9 μm in thickness), dried under vacuum at 40 °C overnight, and then cut into discs (12 mm in diameter) using a disc cutter (MTI). The final composite electrode had a sulfur content of 49 wt %, an active material loading of ~1.0 mg cm⁻², and a thickness of ~64.4 μm. The C-CoNiS/S, MXene/S, and KB/S electrodes were fabricated following the same procedure. Sodium metal electrodes were prepared by rolling sodium metal (Aladdin, 99.7 %) into foils with a thickness of 0.2 mm and a diameter of 12 mm.

### Electrochemical measurements

CR2032 coin cells (Canrd, 304 stainless steel) were assembled in an argon-filled glovebox at room temperature ($H_2O$ and $O_2$ < 0.01 ppm). Each cell was filled with 180 μL of electrolyte (1.0 M $NaSO_3CF_3$ in diethylene glycol dimethyl ether (DEGDME), Dodochem, 99.9 %), resulting in an E/S ratio of 180 μL mg⁻¹. A glass fiber (Whatman GF/D, 14 mm diameter, thickness 0.67 mm) served as separator. Cells were sealed using a hydraulic press (~80 kg cm⁻²). All the cells were aged at room temperature for 4 h before testing. Cycling tests at −20 °C and 50 °C were carried out in an environmental chamber with negligible temperature variation (± 0.5 °C). Room-temperature cycling tests were performed under ambient laboratory conditions at an average temperature of 23 ± 2 °C.

The coin cells were activated for three cycles at specific currents of 0.5 A g⁻¹ within the voltage range of 0.2–2.8 V prior to the main measurements. Galvanostatic charge−discharge tests were conducted on a LAND system. EIS measurements were performed using a CHI 660E electrochemical workstation (Chenhua, Shanghai) in a potentiostatic mode with an AC amplitude of 5 mV over a frequency range of 100 kHz to 0.01 Hz (12 points per decade). For in situ EIS measurements, the cells were galvanostatically charged and discharged at a specific current of 300 mA g⁻¹. EIS spectra were recorded at 0.2 V intervals, with each measurement preceded by a 30 min constant-voltage hold to reach a quasi-equilibrium state. The DRT results were subsequently acquired by transforming the EIS data from frequency to time domain through the MATLAB Graphical User Interface (GUI) toolbox. CV measurements were performed at a scan rate of 0.2 mV s⁻¹ within 0.2–2.8 V vs $Na^+$/Na. For all cycling performance figures, capacity retention and decay rates were calculated using the discharge capacities of the first cycle and the final cycle shown in each figure.

### Polysulfide adsorption tests

2.5 mM $Na_2S_6$ solution was prepared by dissolving stoichiometric amounts of sulfur and anhydrous $Na_2S$ (Aladdin, 95 %) in 10 mL of DEGDME. The mixture was heated to 80 °C and stirred continuously for 48 h in an argon-filled glovebox. For adsorption experiments, 10 mg of powder (A/C-CoNiS or C-CoNiS) was added to 2 mL $Na_2S_6$ solution. After standing for 12 h, the color change of the solution was recorded with a digital camera, and the supernatant was analyzed by UV−vis spectroscopy (after 10-fold dilution). The powders were then dried and characterized by XPS to investigate chemical interactions between the catalysts and polysulfides. To quantify adsorption capacity, UV−vis spectra of $Na_2S_6$ solutions at varying concentrations (0.05, 0.10, 0.15, 0.20, and 0.25 mM) were measured.

### Assembly and testing of symmetric cells

A/C-CoNiS and C-CoNiS powders were separately mixed with PAA at a weight ratio of 9:1 to form a slurry, which was then coated onto copper foil as electrode. Each cell used 40 μL of electrolyte containing $Na_2S_6$ at positive electrode, 40 μL of electrolyte without $Na_2S_6$ at negative electrode, and glass fiber membrane as the separator. CV curves were recorded at a scan rate of 5 mV s⁻¹ within a voltage window of −1.0–1.0 V.

## Theoretical calculations

All first-principles calculations were conducted using the Vienna Ab initio Simulation Package (VASP). The electron exchange-correlation energy was described using the Perdew–Burke–Ernzerhof (PBE) exchange-correlation functional within the generalized gradient approximation (GGA) framework. To accurately describe the strong electron correlation effects in transition metals, the DFT + U method was employed with Hubbard U parameters of 3.32 eV for Co, 6.20 eV for Ni, and 5.80 eV for Ti[52,53]. The empirical dispersions of Grimme (DFT-D3) were applied to account for the long-range van der Waals interactions[49]. To align with experimental characterizations, the (311) surface was cleaved from the $(Co_{0.5}Ni_{0.5})_9S_8$ structure. For calculations involving $Na_2S_x$ (x = 8, 6, 4, 2, and 1), a $1\times2\times1$ Monkhorst-Pack Gamma-centered k-point mesh was employed. The cutoff energy was set to 500 eV for all calculations, with convergence criteria of 0.01 eV Å$^{-1}$ for forces and $10^{-5}$ eV for energy. The adsorption energy ($E_{ads}$) was calculated using the following equation, $E_{ads} = E_{total} - E_{surf} - E_{adsorb}$ where $E_{total}$ represents the total energy of the adsorbed system, $E_{surf}$ is the energy of the optimized clean substrate, and $E_{adsorb}$ denotes the energy of the adsorbate in vacuum. The optimized atomic coordinates are provided in Supplementary Data 1.

## Data availability

The data generated in this study are provided in the Supplementary Information and Source Data file. Source data are provided with this paper.

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

## Acknowledgements

This work was supported by the National Key Research and Development Program of China (Grant No. 2022YFB3807200 to W.Z.), the Fundamental Research Funds for the Central Universities (to W.Z.), and the China Scholarship Council (to W.Z. for an academic visit to Loughborough University).

## Author contributions

T.X. synthesized the samples, performed battery tests, analyzed the experimental results, and wrote the manuscript. Z.F. and J.W. conducted microscopy imaging and TEM analysis. N.R. and J.L. carried out DFT calculations and contributed to theoretical analysis. R.L. assisted with sample synthesis. Y.G. assisted with the in situ EIS measurements. W.L. and H.W. commented on the manuscript. W.Z. conceived the idea, supervised the research, and provided guidance. All authors participated in discussions, provided feedback, and approved the final version. T.X., Z.F., and N.R. contributed equally to this work.

## Competing interests

The authors declare no competing interests.
