## [Transparent Peer Review file · Nature Communications]

Amorphous/crystalline interwoven multipods with high Co/Ni activity for wide-temperature-range sodium-sulfur batteries

Corresponding Author: Professor Wei Zhou

Version 0:

Reviewer comments:

Reviewer #1

(Remarks to the Author)

This manuscript presents a novel catalyst designed for wide-temperature-range sodium-sulfur (Na-S) batteries, demonstrating high specific capacities. The unique amorphous/crystalline interwoven structure significantly enhances the kinetics of the sulfur electrode, particularly in facilitating the complex 16-electron transfer process. The development of wide temperature, especially low-temperature, Na-S batteries is a compelling advancement in the field. A minor revision is recommended, and the following points should be addressed:

Q1. The charge/discharge process involves multi-step reactions. It is crucial for the authors to clearly elucidate the distinct electrochemical pathways influenced by the catalyst. Clarifying how the catalyst modulates these pathways would significantly enhance the understanding of its role in the reaction mechanism.

Q2. Given the inherent thermal instability of sulfur, did the sulfur electrode exhibit any signs of fuming or degradation during the high-temperature tests? Addressing this concern would provide valuable insights into the practical stability and safety of the electrode under operational conditions.

Q3. Please provide detailed coin cell assembly parameters, including but not limited to the compression pressure applied during electrode stacking, the aging time between cell assembly and electrochemical testing, and environmental controls such as temperature during assembly.

Q4. Please add the details of the loading amount of active material of the electrode, the effective area of the current collector and the mass fraction of active components. These parameters are essential for accurately evaluating electrochemical performance.

Q5. Although DFT calculations confirm MXene's adsorption ability, experiment evidence is also needed. Please provide experiment results to strengthen the reliability of the computational findings.

Q6. It looks like the samples correspond to wrong data in Figure 4c. As described in the manuscript, A/C-CoNiS exhibits lower Rct. Please carefully check the data and make necessary change.

Q7. Several figure corrections are required to ensure clarity and consistency.

- The sulfur loading unit in Figure 2c needs to be corrected from "mg cm⁻¹" to "mg cm⁻²."
- The y-axis units in Figure 3a need to be verified for accuracy.
- "ICP-OES" should be correctly spelled.
- It looks a little unclear for the word "crystal" in Figure 1a.
- It's hard to find Sn atoms in Figure 5e.
- There should be no Sn atoms in Figure 5b.

Reviewer #2

(Remarks to the Author)

In this manuscript, CoNiS multipod-like structure with amorphous/crystalline interface were synthesized by the authors through Sn doping approach, which achieved superior performance when applied in Na-S batteries. However, this manuscript not only contains errors in writing and images but also lacks reasonable mechanism and functional explanations for A/C-CoNiS electrodes with ultra-high specific capacity. The authors also did not clearly explain the reasons and characteristics for the excellent electrochemical performance of the material under high and low-temperature conditions. Moreover, the manuscript lacks in-situ characterization methods to support the authors' viewpoint. This manuscript is not

recommended for publication.

1. The specific attribution and meaning of different platforms in the charge-discharge curve (Figure 2a) and different peaks in the CV curve (Figure 4a) have not been explained in detail by the authors.
2. In Figure 4a, the authors mentioned that the oxidation-reduction peak at ~ 0.38 V corresponds to the reduction of the catalyst. Has a new catalyst been formed or not? Does the new catalytic phase play a catalytic role. In-situ electrochemical technology can reveal the dynamic changes of electrode materials during battery charging and discharging processes. In-situ electrochemical research is highly necessary. Please add in-situ electrochemical characterization.
3. The authors mentioned in the manuscript that Sn doping interferes with crystal growth, forming crystalline and amorphous interfaces. What is the uniqueness of Sn? Will doping with other elements have the same effect? The authors should provide relevant experimental data.
4. The sulfur loading on the electrode is an important parameter for battery performance. The authors should provide relevant data.
5. The A/C-CoNiS electrode exhibits relative good electrochemical performance under different temperatures. Please explain in detail the unique advantages and characteristics of this material that can be used as an electrocatalyst for Na-S batteries at different temperatures.
6. The charging and discharging voltage range of the battery in this manuscript is 0.2-2.8 V. Within this voltage range, whether the synthesized sulfide contribute to the capacity or not?
7. In this manuscript, the battery has relative good electrochemical performance, but there are differences and contingencies in electrochemical testing. Please provide three parallel sets of test data and add error bars.
8. The complete cyclic voltammetry curve (charge and discharge) in Figure 3a should be provided by the authors.
9. There is a lack of research summary on amorphous crystalline structures in the field of metal-sulfur batteries. It is recommended to add the latest research results on related materials.
10. There are many type/grammar errors and also images in this manuscript, for example, in line 275, red and green, and there is no green in the image. In Figure 4c, the A/C-CoNiS electrode has a higher electrochemical impedance, and there are errors in plotting and expression.

Reviewer #3

(Remarks to the Author)

The manuscript presents a compelling advancement in Na-S battery catalysts. Strengthening the link between Sn doping, structural modulation, and performance, along with contextualizing claims relative to existing literature, would elevate its impact. Addressing the following issues will ensure broader accessibility. Some comments need to be addressed before publication:

1. Pore size and surface area not clearly stated in the text. Is the structure micro/meso/macroporous?
2. A Whatman glass fiber separator is used instead of a typical Celgard polymer separator. The former is much thicker and will reduce energy density. What is the cycling performance using Celgard separator?
3. There are many reports on sulfur host materials and it is currently known that anode and electrolyte are more of a problem in these battery systems than new cathode host materials. And it is always a bit suspicious if the one and only solution comes up with a new quite exotic composite material.
4. It might be a good idea to quantify the adsorption capacity with UV Vis following the method reported by Hippauf. This would give a good comparison with other more polar carbon host materials as reported by Hao. Also a water isotherm would be helpful to check the hydrophilic character of the host.
5. Of course a big flaw are the limited experimental details. No electrolyte/sulfur ratio given. That's a step backwards in reporting accurate data. With a whatman separator (thick) an excess of electrolyte is guaranteed! The huge capacity loss in the first cycle confirms this view (dissolution). And in a flooded cell PS diffusion takes longer. And the glass fibres also adsorb the PS, and affect crystallization.
It would be good to test the material also with a PE separator and compare it to a Ketjen Black based reference cathode. And check performance also under lean conditions with E/S ratios varying between 5 and 10.
6. Why is cobalt used rather than another metal; is the choice of cobalt important? I don't think the choice of cobalt versus other transition metals is discussed in the paper and the computational authors could test other metal ions!

Reviewer #4

(Remarks to the Author)

This manuscript presents a comprehensive study aimed at mitigating the polysulfide shuttle effect in Na-S batteries through the design and application of a Sn-doped CoNiS catalyst. By integrating both experimental investigations and density functional theory (DFT) calculations, the authors have made commendable efforts to elucidate the role of the electrocatalyst in enhancing the electrochemical performance of Na-S batteries. While the study offers promising insights and contributes to the development of effective shuttle suppression strategies, several critical aspects require further clarification and elaboration. In particular, addressing certain mechanistic details and expanding the discussion of computational findings will significantly strengthen the depth and impact of the work. Specific comments and suggestions are provided below to help improve the overall quality and completeness of the manuscript

1. In this study involving Na_2S_x species ($x = 8, 6, 4, 2,$ and 1), the authors mention the use of a $1 \times 1 \times 1$ Monkhorst-Pack Gamma-centered k-point mesh and a plane-wave energy cutoff of 400 eV for all

calculations. Could the authors please clarify the rationale behind selecting the Gamma point for k-point sampling and the specific cutoff energy value? Additionally, did the authors perform any convergence tests with respect to the k-point mesh and energy cutoff to ensure the reliability and accuracy of the computed results?

2. Considering the presence of transition metal atoms with localized d-electrons in your system, did the authors consider employing the DFT+U approach to better account for on-site Coulomb interactions? If not, could you please elaborate on the justification for omitting the Hubbard U correction, and whether any test calculations were conducted to assess its potential impact on the electronic structure and total energy?

3. In the DFT investigation of the A/C-CoNiS interface and its role in catalytic reaction kinetics, how did the authors determine the specific active site(s) for Na-S adsorption, particularly considering the structural heterogeneity and potentially numerous adsorption sites present in the amorphous region? Given that amorphous structures can host a variety of chemically distinct sites, was any systematic approach like energy screening was employed to identify the most representative or catalytically relevant sites?

4. (a) The results indicate that the amorphous region of the A/C-CoNiS system exhibits the highest binding energies for Na-S adsorption compared to the crystalline counterpart. Could the authors provide further insights into the underlying factors responsible for these enhanced adsorption energies at the amorphous sites? A detailed discussion on the bonding nature—such as charge transfer characteristics, local coordination environments, or electronic structure analysis—would greatly aid in understanding the origin of the stronger interaction.

(b) Was any assessment of the system's electronic conductivity performed to evaluate its influence on the overall catalytic performance? Given that electronic transport plays a crucial role in redox kinetics, such analysis would be of significant value.

(c) In addition, the manuscript briefly mentions Sn doping in the catalyst. Could the authors elaborate on the specific role of Sn in modifying the catalyst's properties? In particular, does Sn incorporation induce any notable changes in the electronic structure or charge redistribution that could influence Na-S adsorption or reaction kinetics?

5. The reversibility of the Na-S battery is a critical parameter influencing long-term performance. In this context, could the authors provide insights into the sodium diffusion mechanism during the charging process? Specifically, how does the Na-S bond cleavage proceed on the catalyst surface, and what are the associated kinetic barriers? It would be helpful if the authors could support their discussion with a kinetic analysis—such as nudged elastic band (NEB) calculations—to quantify the energy barriers involved in Na diffusion and bond dissociation. Was such an analysis performed or considered?

6. MXene has been previously reported to strongly adsorb Na-S species, which may result in structural integrity challenges in Na-S battery systems. Given its role as a substrate in your catalyst design, did the authors consider or evaluate the potential charge transfer between the MXene and the active catalytic components? Such electronic interactions could significantly influence the adsorption strength, electronic structure, and catalytic activity. An analysis of charge redistribution (e.g., Bader charge analysis or density of states) would provide valuable insights into the impact of the MXene support on the overall catalytic behavior.

Version 1:

Reviewer comments:

Reviewer #1

(Remarks to the Author)

The revised manuscript has well addressed my questions. I recommend it for publication pending the following supplementary information:

1. Does Sn doping induce morphological changes in CoNiS? Comparative SEM analysis before and after doping is required.

2. What is the catalyst-to-cathode ratio? Is this the optimal ratio? Additional electrochemical performance data with varying catalyst ratios should be provided.

Reviewer #2

(Remarks to the Author)

The issues I concerned have been addressed by the authors, and the overall quality of the manuscript has been obviously improved. In my opinion, the current manuscript can be accepted for publication on Nature Communications now.

Reviewer #3

(Remarks to the Author)

The author has addressed all my concerns; the manuscript is now acceptable for publication.

Reviewer #4

(Remarks to the Author)

The authors have satisfactorily addressed all comments. I recommend the manuscript be accepted for publication in its current form

Response Letter for Manuscript “Amorphous-crystalline interwoven multipods with high Co/Ni activity for sodium-sulfur battery operated from –20 to 50 °C”

Thank you for reading this letter. We are grateful to the editor for the opportunity to revise our manuscript and wish to extend our sincere thanks to the reviewers for their insightful comments. We have carefully considered all the feedback and revised the manuscript accordingly.

The revision includes five main updates. (1) **Kinetic mechanism**: in situ EIS and corresponding DRT analysis were added to investigate the kinetics of the polysulfide conversion process at 25 and –20 °C. (2) **Adsorption validation**: quantitative adsorption experiments were conducted at 25 and 50 °C to confirm the superior adsorption of A/C-CoNiS for polysulfides. (3) **DFT calculations**: systematic screening of the active sites and analysis of the charge redistribution induced by Sn doping were performed. Supplementary CI-NEB and differential charge density analyses further confirmed the enhanced reaction kinetics and polysulfide adsorption. (4) **Battery configuration and reproducibility**: detailed cell assembly parameters and three independent parallel tests for battery performance were provided. (5) **Figures and clarity**: writing and figure errors were corrected, and some figures as well as supplementary materials were reorganized to improve readability and accuracy.

During the revision, **the author list was updated to reflect contributions**: Nian Ran was added as co-first author for DFT calculations and method improvements (data in this part were all re-calculated and improved); Yuxuan Gao was added for supporting in situ EIS measurements. Xingyu Li was removed due to insufficient contribution. All changes were agreed upon by the original and added authors, with informed consent obtained. A signed author-list-change-form is attached. We appreciate your understanding, and please do not hesitate to contact us if any additional information or materials are required.

Reviewer #1 (Remarks to the Author):

General Comments: *This manuscript presents a novel catalyst designed for wide-temperature-range sodium-sulfur (Na-S) batteries, demonstrating high specific capacities. The unique amorphous/crystalline interwoven structure significantly enhances the kinetics of the sulfur electrode, particularly in facilitating the complex 16-electron transfer process. The development of wide temperature, especially low-temperature, Na-S batteries is a compelling advancement in the field. A minor revision is recommended, and the following points should be addressed:*

Response: Thank you very much for your valuable comments and suggestions, which are helpful to greatly improve the quality of our manuscript. We respond to your comments one by one as follows. Thanks again.

Comment 1-1: *The charge/discharge process involves multi-step reactions. It is crucial for the authors to clearly elucidate the distinct electrochemical pathways influenced by the catalyst. Clarifying how the catalyst modulates these pathways would significantly enhance the understanding of its role in the reaction mechanism.*

Response 1-1: We sincerely thank the reviewer for the insightful comment. We separately discuss the roles of the catalyst in the discharge process steps of (1) Na_2S_x ($6 < x \leq 8$) \rightarrow Na_2S_4 , (2) $\text{Na}_2\text{S}_4 \rightarrow \text{Na}_2\text{S}$, as well as (3) in the charge process.

(1) Discharge process: Na_2S_x ($6 < x \leq 8$) \rightarrow Na_2S_4

Long-chain soluble polysulfides induce severe shuttle effects. Quantitative UV-vis analysis reveals that A/C-CoNiS has a significantly higher Na_2S_6 adsorption capacity than C-CoNiS, particularly at 50 °C (Supplementary Table 3, Fig. 4h and Supplementary Fig.21 and 22). As a result, the shuttle effect is suppressed, which is beneficial to the cycling performance of the battery.

(2) Discharge process: $\text{Na}_2\text{S}_4 \rightarrow \text{Na}_2\text{S}$

In situ EIS tests were conducted to investigate the reaction kinetics of the battery, especially during the rate-limiting step. In this process, the charge transfer resistance (R_{ct}) of A/C-CoNiS/S remains stable, whereas the R_{ct} of C-CoNiS/S exhibits a significant increase. This indicates that A/C-CoNiS/S has an excellent catalytic effect on the rate-determining step (Supplementary Fig. 25). Under low-temperature conditions, the difference between the two sample is even more pronounced. The according Distribution of Relaxation Times (DRT) profiles in Fig. 4c-f further confirm this conclusion.

(3) Charge process

The ion diffusion significantly slows down at low temperatures. As a result, in the DRT spectra of A/C-CoNiS/S and C-CoNiS/S at $-20\text{ }^\circ\text{C}$ in Fig. 4c-f, there is a pronounced peak during the charge process in the relaxation time around 10^{-2} s, which corresponds to the ion diffusion process (*J. Am. Chem. Soc.* **2025**, *147*, 8250). The significantly reduced signal indicates that A/C-CoNiS can significantly accelerate the ion diffusion. This enables the battery to maintain excellent rate performance even at low temperatures. Additionally, the smaller R_{ct} also indicates A/C-CoNiS/S exhibits faster reaction kinetics during the charging process.

In summary, during the three discharge/charge processes discussed above, the main roles of A/C-CoNiS are to suppress the shuttle effect of polysulfides, accelerate the reaction kinetics of the rate-determining step, and promote the ion diffusion, respectively. These features contribute to the superior electrochemical performance across a wide temperature range of A/C-CoNiS/S. We have further expanded the mechanistic discussion in the revised manuscript with detailed analyses of each conversion step.

Related Revision:

(Manuscript, page 11–12, 28)

Fig. 4 | Enhanced kinetic rate and adsorption ability confirmed by experiments. a CV curves of A/C-CoNiS/S and C-CoNiS/S at 0.2 mV s^{-1} at $0.2\sim 2.8 \text{ V vs Na}^+/\text{Na}$. **b** CV curves of symmetric cells using A/C-CoNiS and C-CoNiS. **DRT plots of c** A/C-CoNiS/S ($25 \text{ }^\circ\text{C}$), **d** C-CoNiS/S ($25 \text{ }^\circ\text{C}$), **e** A/C-CoNiS/S ($-20 \text{ }^\circ\text{C}$), and **f** C-CoNiS/S ($-20 \text{ }^\circ\text{C}$) obtained from in situ EIS measurements. **g** Binding energies of S_8 and NaPSs on MXene. **h** UV-vis spectra of Na_2S_6 solutions absorbed by A/C-CoNiS and C-CoNiS with optical photos inserted. **i** High-resolution Co 2p XPS spectrums of A/C-CoNiS before and after adsorbing Na_2S_6 .

In situ EIS tests and corresponding distribution of relaxation times (DRT) analyses were conducted at 25 and $-20 \text{ }^\circ\text{C}$ to evaluate the electrochemical catalytic performance across the full working voltage range (Figs. 4c-f, Supplementary Fig. 23). The time constant (τ) represents

the relaxation time of each electrochemical process. As shown in Figs. 4c-f, each contour plot can be divided into four regions corresponding to interphase contact resistance (R_1), charge transfer resistance (R_2 , R_{ct}), polysulfide diffusion resistance (R_3), and ion diffusion resistance (R_4)³⁴. At 25 °C, the A/C-CoNiS/S electrode consistently exhibits lower R_{ct} than C-CoNiS/S (Figs. 4c and 4d). This trend becomes more pronounced at -20 °C (Figs. 4e and 4f), indicating the excellent catalytic performance of A/C-CoNiS/S at low temperatures.

(Manuscript, page 14)

Moreover, in situ EIS and corresponding DRT analysis verify that A/C-CoNiS effectively accelerates reaction kinetics particularly under low-temperature conditions.

(Supplementary information, page 27)

Supplementary Fig. 23 | In situ EIS of Na-S batteries with different cathodes under various temperatures. a A/C-CoNiS/S at 25 °C, **b** C-CoNiS/S at 25 °C, **c** A/C-CoNiS/S at -20 °C, and **d** C-CoNiS/S at -20 °C. **e, f** Full-range impedance spectra corresponding to the representative curves at 1.6 and 1.8 V during discharge process in **c** and **d**, respectively. In situ EIS was employed to monitor the reaction kinetics during discharge/charge process. The R_{ct} of A/C-CoNiS/S remains consistently lower than that of C-CoNiS/S at 25 °C by observing semicircle diameter. The difference between the two samples becomes more pronounced at

–20 °C. Furthermore, the maximum value of R_{ct} for both A/C-CoNiS/S and C-CoNiS/S can be detected at ~1.8 V at 25 °C. However, at –20 °C, the peak R_{ct} of A/C-CoNiS/S remains at 1.8 V with a slight increase, while that of C-CoNiS/S shifts to 1.6 V and rises sharply, suggesting enhanced polarization and slower kinetics^{34,35}. The results indicate A/C-CoNiS/S maintains stable charge transfer kinetics even at low temperature, demonstrating a superior low-temperature adaptability³⁵.

(Manuscript, page 12)

Fig. 4h shows that A/C-CoNiS exhibits a markedly weaker UV-vis absorption signal than C-CoNiS and the Na_2S_6 solution at room temperature, revealing superior adsorption ability. Quantitative UV-vis adsorption tests (Supplementary Figs. 24 and 25, Table 4) confirm that A/C-CoNiS exhibits superior adsorption capacities especially at high temperatures, indicating an improved anti-shuttle effect.

(Supplementary information, page 28–30)

Supplementary Fig. 24 | A linear calibration curve was established based on Na_2S_6 solutions of known concentrations. a UV-vis absorption spectra of Na_2S_6 solutions with varying concentrations. **b** Corresponding calibration curve used for quantitative analysis. Based on the established calibration curve, the adsorption capacity of the catalyst for polysulfides can be quantitatively determined from the corresponding UV-vis absorption data.

Supplementary Fig. 25 | UV-vis spectra of Na_2S_6 solutions after immersion with A/C-CoNiS and C-CoNiS at 50 °C for 12 h. Insets: digital photographs of the solutions before and after adsorption. Compared to C-CoNiS and the Na_2S_6 solution, A/C-CoNiS exhibits weaker UV-vis signals and a more transparent solution (inset), indicating stronger NaPSs adsorption and improved anti-shuttle capability ⁴.

Supplementary Table 4.

Polysulfide adsorption capacity of A/C-CoNiS and C-CoNiS at different temperatures.

Sample	Temperature (°C)	Adsorption Capacity (mmol·g ⁻¹)	Adsorption Capacity (g _s ·g ⁻¹)
A/C-CoNiS	25	0.336	0.080
	50	0.328	0.078
C-CoNiS	25	0.235	0.056
	50	0.086	0.020

(Manuscript, page 18)

Polysulfide adsorption tests

2.5 mM Na_2S_6 solution was prepared by dissolving stoichiometric amounts of sulfur and anhydrous sodium sulfide (Na_2S , AR) in 10 mL of DEGDME. The mixture was heated to 80 °C and stirred continuously for 48h in an argon-filled glove box. For adsorption experiments, 10 mg of powder (A/C-CoNiS or C-CoNiS) was added to 2 mL Na_2S_6 solution. After standing

for 12h, the color change of the solution was recorded with a digital camera, and the supernatant was analyzed by UV-vis spectroscopy (after 10-fold dilution). The powders were then dried and characterized by XPS to investigate chemical interactions between the catalysts and polysulfides. To quantify adsorption capacity, UV-vis spectra of Na₂S₆ solutions at varying concentrations (0.05, 0.10, 0.15, 0.20, and 0.25 mM) were measured.

Comment 1-2: *Given the inherent thermal instability of sulfur, did the sulfur electrode exhibit any signs of fuming or degradation during the high-temperature tests? Addressing this concern would provide valuable insights into the practical stability and safety of the electrode under operational conditions.*

Response 1-2: We appreciate the reviewer's comment. TGA results (Supplementary Fig. 11) show noticeable mass loss only above 55 °C. Since 50 °C is well below the boiling point of elemental sulfur (~115 °C) (*Natl. Sci. Rev.* **2023**, *10*, 268), significant evaporation is unlikely at this temperature. Moreover, long-term cycling at 50 °C shows a capacity decay of only 0.002% per cycle (Fig. 3e), further confirming that sulfur loss is negligible. In summary, these results demonstrate that the sulfur electrode exhibits excellent thermal stability and good safety at 50 °C.

Related Revision:

(Supplementary information, page 13)

Supplementary Fig. 11 | TGA profiles of A/C-CoNiS/S and C-CoNiS/S. TGA tests were conducted over the temperature range of 30 to 400 °C with a heating rate of 10 °C min⁻¹. No significant mass loss was observed before 55 °C. The rapid weight loss observed between 150 and 350 °C is attributed to the volatilization of sulfur. The sulfur content in A/C-CoNiS/S is ~70 wt%, which is close to the sulfur content in the counterpart of C-CoNiS/S (~65 wt%).

Comment 1-3: *Please provide detailed coin cell assembly parameters, including but not limited to the compression pressure applied during electrode stacking, the aging time between cell assembly and electrochemical testing, and environmental controls such as temperature during assembly.*

Response 1-3: We thank the reviewer for the insightful suggestion. As shown below, detailed assembly parameters have been included in the revised Methods section of the manuscript.

Related Revision:

(Manuscript, page 17)

Electrochemical measurements

CR2032 coin cells were assembled in an argon-filled glovebox at room temperature (H₂O and O₂ < 0.01 ppm). Each cell was filled with 180 μL of 1.0 M NaSO₃CF₃ in DEGDME. A Whatman glass fiber membrane and a sodium foil (1 mm thick, 14 mm diameter) served as separator and counter electrode, respectively. Cells were sealed using a hydraulic press (~80 kg cm⁻²). All the cells were aged at room temperature for 4h before testing. Galvanostatic charge–discharge tests were conducted between 0.2–2.8 V on a LAND system. CV was conducted at the same voltage range, and EIS was performed at open-circuit potential across a frequency range of 0.01~100 kHz using a CHI600E electrochemical workstation. **In situ EIS** was carried out in the same frequency range during galvanostatic charge–discharge cycling with an interval of 30 min. The DRT results were subsequently acquired by transforming the

EIS data from frequency to time domain through the MATLAB Graphical user interface (GUI) toolbox.

Comment 1-4: *Please add the details of the loading amount of active material of the electrode, the effective area of the current collector and the mass fraction of active components. These parameters are essential for accurately evaluating electrochemical performance.*

Response 1-4: Thank you for this valuable suggestion. According to the reviewers' comments, we have added relevant information in the revised manuscript.

Related Revision:

(Manuscript, page 16)

Preparation of sulfur cathodes

The sulfur and A/C-CoNiS powders were mixed with a mass ratio of 7:3 for 10 min in an agate mortar. The mixture was transferred to a stainless-steel high-pressure reactor, sealed under argon in a glove box, and heated at 155 °C for 12h to allow sulfur diffusion into the A/C-CoNiS. The working electrodes were prepared by mixing active material, carbon black, and PAA binder (7:2:1 by weight) in deionized water, casting the obtained slurry onto copper foil, and drying under vacuum at 40 °C overnight. The active material loading was $\sim 1.0 \text{ mg cm}^{-2}$ with an electrode area of 1.13 cm^2 . The C-CoNiS/S, MXene/S, and KB/S cathodes were fabricated by the same way.

Comment 1-5: *Although DFT calculations confirm MXene's adsorption ability, experiment evidence is also needed. Please provide experiment results to strengthen the reliability of the computational findings.*

Response 1-5: We thank the reviewer for this valuable suggestion. To experimentally validate the adsorption capability of MXene, we conducted UV-vis adsorption measurements. As

shown in Fig. R1, the characteristic peak of Na₂S₆ was significantly weakened after adsorption. In addition, the adsorption capacity of MXene toward Na₂S₆ was quantitatively calculated to be 0.41 mmol g⁻¹ based on the results displayed in the Supplementary Fig. 24, which further confirms its strong affinity toward polysulfides and thus supports the computational findings.

Fig. R1 UV-vis spectra of Na₂S₆ solutions after immersion with MXene at 25 °C for 12 h.

Comment 1-6: *It looks like the samples correspond to wrong data in Figure 4c. As described in the manuscript, A/C-CoNiS exhibits lower R_{ct}. Please carefully check the data and make necessary change.*

Response 1-6: We appreciate the reviewer’s careful observation. We apologize for the oversight and correct this mistake in the revised manuscript.

Related Revision:

(Supplementary information, page 31)

Supplementary Fig. 22 | Nyquist plots and corresponding R_{ct} values with the equivalent circuit model inserted. The R_{ct} value of A/C-CoNiS/S is 8.1 Ω , much lower than that of C-CoNiS/S (19.8 Ω), indicating its faster charge transfer.

Comment 1-7: *Several figure corrections are required to ensure clarity and consistency.*

a. The sulfur loading unit in Figure 2c needs to be corrected from “mg cm⁻¹” to “mg cm⁻².”

b. The y-axis units in Figure 3a need to be verified for accuracy.

c. “ICP-OES” should be correctly spelled.

d. It looks a little unclear for the word “crystal” in Figure 1a.

e. It’s hard to find Sn atoms in Figure 5e.

f. There should be no Sn atoms in Figure 5b.

Response 1-7: We thank the reviewer for the helpful suggestions. The necessary corrections have been made, and the related figures have been carefully checked and revised accordingly to ensure consistency and clarity. Due to the large number of changes, we do not list each one here, but all revisions have been highlighted in the revised manuscript and SI.

Reviewer #2 (Remarks to the Author):

General Comments: *In this manuscript, CoNiS multipod-like structure with amorphous/crystalline interface were synthesized by the authors through Sn doping approach, which achieved superior performance when applied in Na-S batteries. However, this manuscript not only contains errors in writing and images but also lacks reasonable mechanism and functional explanations for A/C-CoNiS electrodes with ultra-high specific capacity. The authors alao did not clearly explain the reasons and characteristics for the excellent electrochemical performance of the material under high and low-temperature conditions. Moreover, the manuscript lacks in-situ characterization methods to support the authors' viewpoint. This manuscript is not recommended for publication.*

Response: We sincerely appreciate your thorough and meticulous evaluation of our manuscript, as well as the insightful and constructive comments provided. Based on your comments, we have made the corresponding revisions. We have carefully revised the entire manuscript to ensure that both the text and figures are free of errors. Building on the existing results and supplementary testing (in-situ EIS measurements, quantitative adsorption tests, etc.), together with support from the literature, we have conducted an in-depth investigation. These efforts have enabled us to systematically elucidate the origin of the high specific capacity of A/C-CoNiS/S, uncover the regulatory role of A/C-CoNiS in the reaction process, and clarify the mechanisms underlying its outstanding electrochemical performance across a wide temperature range. We response to your comments one by one as following. Thanks again.

Comment 2-1: *The specific attribution and meaning of different platforms in the charge-discharge curve (Figure 2a) and different peaks in the CV curve (Figure 4a) have not been explained in detail by the authors.*

Response 2-1: We sincerely thank the reviewer for the insightful comment, which helps improve the clarity and depth of our electrochemical analysis. In response, we provide a detailed interpretation of the charge-discharge plateaus observed in Fig. 2a and the corresponding redox peaks in the CV curves shown in Fig. 4a. The revised explanations are presented in the revised manuscript.

Related Revision:

(Manuscript, page 6)

Fig. 2a shows the galvanostatic charge-discharge (GCD) profiles of A/C-CoNiS/S and C-CoNiS/S electrodes. The discharge curve of A/C-CoNiS/S exhibits three voltage plateaus at ~2.0, 1.5, and 0.8 V, corresponding to the conversion from S₈ to long-chain polysulfides (Na₂S_x, 6<x≤8), then to Na₂S₄, and finally to Na₂S. Upon charging, two main plateaus appear at 1.7 and 2.2 V, corresponding to the stepwise oxidation of Na₂S to Na₂S₄ and then to Na₂S_x²⁸.

(Manuscript, page 10–11)

The CV curves in Fig. 4a display a series of well-defined reduction and oxidation peaks. Four distinct reduction peaks are observed at 1.97 (R₀), 1.53 (R₁), 1.17 (R₂), and 0.81 V (R₃), corresponding to the stepwise conversion of S₈ to Na₂S_x (R₀), Na₂S_x to Na₂S₄ (R₁), and Na₂S₄ to Na₂S (R₂, R₃)^{16,36}. In addition, two peaks at 0.94 V (R_{M1}) and 0.50 V (R_{M2}) are ascribed to Na⁺ insertion into CoNiS⁴⁶. The oxidation peaks at 1.30 (O₁, M₁), 1.54 (O₂), 1.72 (O₃, M₂), and 2.15 V (O₄) correspond to the reverse process accordingly³⁴.

The evolution of the CV curves during the initial cycles is shown in Supplementary Fig. 21.

Comment 2-2: *In Figure 4a, the authors mentioned that the oxidation-reduction peak at ~0.38 V corresponds to the reduction of the catalyst. Has a new catalyst been formed or not? Does the new catalytic phase play a catalytic role. In-situ electrochemical technology can reveal the dynamic changes of electrode materials during battery charging and discharging processes.*

In-situ electrochemical research is highly necessary. Please add in-situ electrochemical characterization.

Response 2-2: We appreciate the reviewer's insightful question. Based on the CV and XRD results shown below, we confirm that no new catalyst was formed at this potential. We performed CV analysis of a Na-ion battery (A/C-CoNiS as the anode) and a Na-S battery (A/C-CoNiS/S). The results show that the Na-ion battery exhibits no reduction peak near 0.38 V (Fig. R2a), whereas the A/C-CoNiS/S battery displays a ~0.38 V reduction peak during the first three CV cycles, with gradually decreasing intensity until it disappears in the fourth cycle (Supplementary Fig. 23). The peak is most likely attributable to the formation of the SEI layer in the early cycles (*ACS Nano* **2023**, *17*, 19063-19075; *Adv Mater* **2020**, *32*, e1906700), rather than to catalyst reduction or the emergence of a new phase. Furthermore, the XRD pattern of A/C-CoNiS/S at 0.2 V (Fig. R2b) reveals no phase transformation. Taken together, the results indicate that no new catalytic phase is generated during cycling. The relevant content has been revised in manuscript and SI.

We also fully agree with the suggestion on the necessity of in-situ electrochemical characterization. Accordingly, we have performed in-situ EIS measurements to probe the dynamic behavior of the electrode during cycling at both 25 °C and -20 °C. Please check the corresponding revisions in the manuscript and SI.

Fig. R2 | **a** CV curve of the Na-ion battery using A/C-CoNiS as the anode. **b** XRD pattern of the A/C-CoNiS/S electrode discharged to 0.2 V.

Related Revision:

(Supplementary information, page 25)

Supplementary Fig. 21 | CV curves of A/C-CoNiS/S at 0.2 mV s^{-1} . During the initial cathodic scan, the reduction peaks at 2.15, 1.23, and 0.81 V correspond to the conversion of S₈ to long-chain polysulfides (Na₂S_x, $6 < x \leq 8$), the transformation of Na₂S_x to short-chain Na₂S₄, and the final reduction of Na₂S₄ to Na₂S, respectively^{9,28-30}. In addition, the reduction peak at approximately 0.23 V is most likely attributable to the formation of the SEI layer^{31,32}. The oxidation peaks are associated with the stepwise oxidation of Na₂S to Na₂S_x²³. In the CV curves of the later cycles, the reduction peaks related to sulfur species exhibit a noticeable shift and tend to stabilize, while the reduction peaks associated with the SEI gradually weaken and disappear by the fourth cycle^{31,32}. Furthermore, the oxidation peaks split from a single peak into multiple peaks and eventually stabilize. The splitting of the oxidation peaks may be due to more thorough oxidation of Na₂S³³. To investigate the properties of the battery during stable cycling, the CV data were collected from the fourth cycle unless otherwise specified.

(Manuscript, page 11–12, 28)

Fig. 4 | Enhanced kinetic rate and adsorption ability confirmed by experiments. **a** CV curves of A/C-CoNiS/S and C-CoNiS/S at 0.2 mV s^{-1} at $0.2\sim 2.8 \text{ V vs Na}^+/\text{Na}$. **b** CV curves of symmetric cells using A/C-CoNiS and C-CoNiS. **c** DRT plots of A/C-CoNiS/S (25°C), **d** C-CoNiS/S (25°C), **e** A/C-CoNiS/S (-20°C), and **f** C-CoNiS/S (-20°C) obtained from in situ EIS measurements. **g** Binding energies of S_8 and NaPSs on MXene. **h** UV-vis spectra of Na_2S_6 solutions absorbed by A/C-CoNiS and C-CoNiS with optical photos inserted. **i** High-resolution Co 2p XPS spectrums of A/C-CoNiS before and after adsorbing Na_2S_6 .

In situ EIS tests and corresponding distribution of relaxation times (DRT) analyses were conducted at 25 and -20°C to evaluate the electrochemical catalytic performance across the full working voltage range (Figs. 4c-f, Supplementary Fig. 23). The time constant (τ) represents

the relaxation time of each electrochemical process. As shown in Figs. 4c-f, each contour plot can be divided into four regions corresponding to interphase contact resistance (R1), charge transfer resistance (R2, R_{ct}), polysulfide diffusion resistance (R3), and ion diffusion resistance (R4)³⁴. At 25 °C, the A/C-CoNiS/S electrode consistently exhibits lower R_{ct} than C-CoNiS/S (Figs. 4c and 4d). This trend becomes more pronounced at -20 °C (Figs. 4e and 4f), indicating the excellent catalytic performance of A/C-CoNiS/S at low temperatures.

(Manuscript, page 14)

Moreover, in situ EIS and corresponding DRT analysis verify that A/C-CoNiS effectively accelerates reaction kinetics particularly under low-temperature conditions.

(Supplementary information, page 27)

Supplementary Fig. 23 | In situ EIS of Na-S batteries with different cathodes under various temperatures. a A/C-CoNiS/S at 25 °C, **b** C-CoNiS/S at 25 °C, **c** A/C-CoNiS/S at -20 °C, and **d** C-CoNiS/S at -20 °C. **e, f** Full-range impedance spectra corresponding to the representative curves at 1.6 and 1.8 V during discharge process in **c** and **d**, respectively. In situ EIS was employed to monitor the reaction kinetics during discharge/charge process. The R_{ct} of A/C-CoNiS/S remains consistently lower than that of C-CoNiS/S at 25 °C by observing semicircle diameter. The difference between the two samples becomes more pronounced at

–20 °C. Furthermore, the maximum value of R_{ct} for both A/C-CoNiS/S and C-CoNiS/S can be detected at ~1.8 V at 25 °C. However, at –20 °C, the peak R_{ct} of A/C-CoNiS/S remains at 1.8 V with a slight increase, while that of C-CoNiS/S shifts to 1.6 V and rises sharply, suggesting enhanced polarization and slower kinetics^{34,35}. The results indicate A/C-CoNiS/S maintains stable charge transfer kinetics even at low temperature, demonstrating a superior low-temperature adaptability³⁵.

Comment 2-3: The authors mentioned in the manuscript that Sn doping interferes with crystal growth, forming crystalline and amorphous interfaces. What is the uniqueness of Sn? Will doping with other elements have the same effect? The authors should provide relevant experimental data.

Response 2-3: We thank the reviewer for this valuable question. To elucidate the uniqueness of Sn doping, we synthesized a series of control samples with Mn, Cu, and Zn doping under identical conditions (Supplementary Figs. 5–7). The results reveal that only Sn doping enables the formation of well-defined crystalline–amorphous interfaces. Specifically, Mn shows negligible structural impact due to its ultralow doping level; Cu induces global polycrystallization with indistinct amorphous regions; and Zn leads to defect-rich polycrystalline nanoparticles where amorphous regions are hard to distinguish. In contrast, Sn uniquely enables partial disruption of crystal growth while preserving the single-crystalline backbone, resulting in localized amorphous regions and well-defined interfaces (Fig. 1d, Supplementary Fig. 4). This comparative analysis highlights that the atomic-scale interface engineering observed in Sn-doped samples cannot be replicated by other dopants, confirming the critical role of Sn in modulating phase structure and catalytic functionality. According to the reviewers’ comments, we have added a discussion of other metal doping in the revised manuscript and SI.

Related Revision:

(Manuscript, page 6)

Additionally, three other transition metals (Cu, Mn, and Zn) were selected for doping, but no comparable effects were observed (Supplementary Figs. 5-7).

(Supplementary Information, page 7–9)

Supplementary Fig. 5 | Characterizations on Cu-CoNiS. a XRD pattern. b HAADF-STEM

image. c, d HRTEM image. e HAADF-STEM image with elemental mapping. f EDS spectrum.

To explore the effects of metal doping, a Cu-doped CoNiS sample (Cu-CoNiS) was synthesized. XRD pattern confirms the main phase of (CoNi)₉S₈. Supplementary Fig. 5b

shows that Cu doping induces a distinct bowl-like morphology. Multiple randomly oriented

lattice fringes confirm its polycrystalline structure (Supplementary Fig. 5c). The HRTEM

image shows lattice fringes with a spacing of 0.28 nm, which can be indexed to the (311)

planes of (CoNi)₉S₈. Elemental mapping and EDS analysis reveal uniform distribution of Co,

Ni, and Cu with a molar ratio of 1: 0.8: 0.3. Unlike Sn doping, Cu can be readily incorporated,

which significantly alters the nanostructure. Since the nanobowls are polycrystalline, it is

difficult to clearly distinguish between crystalline and amorphous regions, unlike in single-crystalline structures where the interface can be easily identified.

Supplementary Fig. 6 | Characterizations on Mn-CoNiS. **a** XRD pattern. **b** HAADF-STEM image. **c, d** HRTEM image. **e** Elemental mapping. **f** EDS spectrum. To investigate the effect of metal doping, a Mn-doped CoNiS sample (Mn-CoNiS) was synthesized. XRD analysis (Supplementary Fig. 6a) confirms that the main phase remains $(\text{CoNi})_9\text{S}_8$. The sample exhibits a bowl-like morphology, as shown in Supplementary Fig. 6b. Multiple randomly oriented lattice fringes confirm its polycrystalline structure (Supplementary Fig. 6c). HRTEM imaging (Supplementary Fig. 6d) reveals lattice fringes with a spacing of 0.28 nm, corresponding to the (311) plane of $(\text{CoNi})_9\text{S}_8$. Elemental mapping (Supplementary Fig. 6e) and EDS analysis (Supplementary Fig. 6f) reveal a uniform distribution of Co, Ni, and Mn with a molar ratio of 1: 1: 0.01. These results suggest that Mn was not effectively incorporated into the CoNiS lattice. The bowl is also polycrystalline, which is hard to distinguish the crystal and amorphous regions.

Supplementary Fig. 7 | Characterizations on Zn-CoNiS. a XRD pattern. b HAADF-STEM

image. c, d HRTEM image. e HAADF-STEM image with elemental mapping. f EDS spectrum.

To explore the effects of metal doping, a Zn-doped CoNiS sample (Zn-CoNiS) was synthesized. XRD pattern confirms the main phase of $(\text{CoNi})_9\text{S}_8$. Zn doping induces the formation of ~ 10 nm nanoparticles on the MXene surface (Supplementary Fig. 7b). The HRTEM image shows lattice fringes with a spacing of 0.18 nm, which can be indexed to the (440) planes of $(\text{CoNi})_9\text{S}_8$. Elemental mapping and EDS analysis reveal uniform distribution of Co, Ni, and Zn with a molar ratio of 1: 0.9: 0.1. Unlike Sn doping, Zn doping results in nanoparticles with abundant defects. The ultrasmall size makes it difficult to distinguish the amorphous–crystalline interface, in sharp contrast to the clearly discernible amorphous region in the Sn-doped counterpart.

Comment 2-4: *The sulfur loading on the electrode is an important parameter for battery performance. The authors should provide relevant data.*

Response 2-4: We thank the reviewer for highlighting this important aspect. The sulfur loading in all sodium–sulfur battery tests was controlled at $\sim 1.0 \text{ mg cm}^{-2}$. This information has been added in the “Electrochemical measurements” section of the revised manuscript.

Related Revision:

(Manuscript, page 17)

The active material loading was $\sim 1.0 \text{ mg cm}^{-2}$ with an electrode area of 1.13 cm^2 . The C-CoNiS/S, MXene/S, and KB/S cathodes were fabricated by the same way.

Comment 2-5: *The A/C-CoNiS electrode exhibits relative good electrochemical performance under different temperatures. Please explain in detail the unique advantages and characteristics of this material that can be used as an electrocatalyst for Na–S batteries at different temperatures.*

Response 2-5: We sincerely appreciate the reviewer’s valuable comments. At high temperature of $50 \text{ }^\circ\text{C}$, A/C-CoNiS effectively suppresses the shuttle effect of polysulfides, thereby enhancing the cycling stability of the battery. At low temperature of $-20 \text{ }^\circ\text{C}$, A/C-CoNiS effectively catalyzes the transformation of polysulfides, promoting the reaction kinetics of the battery. As a result, the obtained Na–S batteries exhibit excellent electrochemical performance across a wide temperature range. Detailed discussions are provided below.

At $50 \text{ }^\circ\text{C}$, intensified polysulfide diffusion and shuttle effects typically degrade performance. UV–vis absorption quantification demonstrates that A/C-CoNiS maintains stronger and more stable polysulfide adsorption at both room temperature and $50 \text{ }^\circ\text{C}$, effectively mitigating polysulfide migration and thereby enhancing cycling stability (Supplementary Table 3). The significantly improved cycling stability of A/C-CoNiS/S at $50 \text{ }^\circ\text{C}$ in Fig. 3e further confirms this conclusion.

At $-20\text{ }^{\circ}\text{C}$, the reaction kinetics are generally limited by higher activation barriers. CV curves (Fig. 3a) reveal only a slight positive shift ($\sim 0.09\text{ V}$) in the oxidation peak potential for A/C-CoNiS/S compared to a larger shift ($\sim 0.20\text{ V}$) for the control. Furthermore, in situ EIS tests and corresponding DRT plots (Fig. 4c-f and Supplementary Fig. 25) show that the R_{ct} of the A/C-CoNiS/S is significantly lower than that of C-CoNiS/S, indicating higher catalytic activity that promotes reversible polysulfide conversion and accelerates reaction kinetics. The higher capacity and better rate performance of A/C-CoNiS/S at $-20\text{ }^{\circ}\text{C}$ also support this conclusion (Fig. 3b, c).

This discussion has been incorporated into the revised manuscript and SI to emphasize the unique advantages of A/C-CoNiS in Na-S batteries at different temperatures.

Related Revision:

(Manuscript, page 12)

Fig. 4h shows that A/C-CoNiS exhibits a markedly weaker UV-vis absorption signal than C-CoNiS and the Na_2S_6 solution at room temperature, revealing superior adsorption ability. Quantitative UV-vis adsorption tests (Supplementary Figs. 24 and 25, Table 4) confirm that A/C-CoNiS exhibits superior adsorption capacities especially at high temperatures, indicating an improved anti-shuttle effect.

(Supplementary information, page 28–30)

Supplementary Fig. 24 | A linear calibration curve was established based on Na₂S₆ solutions of known concentrations. a UV-vis absorption spectra of Na₂S₆ solutions with varying concentrations. b Corresponding calibration curve used for quantitative analysis. Based on the established calibration curve, the adsorption capacity of the catalyst for polysulfides can be quantitatively determined from the corresponding UV-vis absorption data.

Supplementary Fig. 25 | UV-vis spectra of Na₂S₆ solutions after immersion with A/C-CoNiS and C-CoNiS at 50 °C for 12 h. Insets: digital photographs of the solutions before and after adsorption. Compared to C-CoNiS and the Na₂S₆ solution, A/C-CoNiS exhibits weaker UV-vis signals and a more transparent solution (inset), indicating stronger NaPSs adsorption and improved anti-shuttle capability⁴.

Supplementary Table 4.

Polysulfide adsorption capacity of A/C-CoNiS and C-CoNiS at different temperatures.

Sample	Temperature (°C)	Adsorption Capacity (mmol·g ⁻¹)	Adsorption Capacity (g _s ·g ⁻¹)
A/C-CoNiS	25	0.336	0.080
	50	0.328	0.078
C-CoNiS	25	0.235	0.056
	50	0.086	0.020

(Manuscript, page 18)

Polysulfide adsorption tests

2.5 mM Na₂S₆ solution was prepared by dissolving stoichiometric amounts of sulfur and anhydrous sodium sulfide (Na₂S, AR) in 10 mL of DEGDME. The mixture was heated to 80 °C and stirred continuously for 48h in an argon-filled glove box. For adsorption experiments, 10 mg of powder (A/C-CoNiS or C-CoNiS) was added to 2 mL Na₂S₆ solution. After standing for 12h, the color change of the solution was recorded with a digital camera, and the supernatant was analyzed by UV-vis spectroscopy (after 10-fold dilution). The powders were then dried and characterized by XPS to investigate chemical interactions between the catalysts and polysulfides. To quantify adsorption capacity, UV-vis spectra of Na₂S₆ solutions at varying concentrations (0.05, 0.10, 0.15, 0.20, and 0.25 mM) were measured.

Comment 2-6: *The charging and discharging voltage range of the battery in this manuscript is 0.2-2.8 V. Within this voltage range, whether the synthesized sulfide contribute to the capacity or not?*

Response 2-6: We thank the reviewer for this insightful question. To evaluate the capacity contribution of the synthesized sulfide within the voltage range of 0.2–2.8 V, we conducted cycling tests at a current density of 1 A g⁻¹ (Supplementary Fig. 12b). After 250 cycles, the electrode delivers a capacity of 215.8 mAh g⁻¹. Considering its 30 wt% content in the A/C-CoNiS/S composite, A/C-CoNiS contributes only 6.4% of the total capacity, confirming that sulfur is the primary capacity source.

Related Revision:

(Manuscript, page 7)

Supplementary Fig. 12a shows that the sodium-ion storage performance of A/C-CoNiS contributes minimally to the overall capacity, indirectly indicating that the main contribution comes from the redox process of sulfur. The MXene/S and Ketjen Black/S cathodes deliver

capacities far below A/C-CoNiS/S, highlighting the catalytic advantage of A/C-CoNiS (Supplementary Fig. 12b and 12c).

(Supplementary information, page 16)

Supplementary Fig. 12 | Cycling stability of the A/C-CoNiS, MXene/S, and KB/S at 1 A g⁻¹. In sodium-ion batteries, A/C-CoNiS delivers 215.8 mAh g⁻¹ after 250 cycles at 1 A g⁻¹ (Supplementary Fig. 12a). Considering its 30 wt% content in the A/C-CoNiS/S composite, A/C-CoNiS contributes only 6.4% of the total capacity, confirming that sulfur is the primary capacity source. In Na-S batteries, MXene/S and KB/S electrodes deliver discharge capacities of 365.1 and 339.4 mAh g⁻¹ respectively after 250 cycles at 1 A g⁻¹ (Supplementary Figs. 12b, c), highlighting the superior catalytic performance of A/C-CoNiS.

Comment 2-7: *In this manuscript, the battery has relative good electrochemical performance, but there are differences and contingencies in electrochemical testing. Please provide three parallel sets of test data and add error bars.*

Response 2-7: We sincerely appreciate the reviewer's valuable suggestion. The data at each temperature are presented as mean values with standard deviations (mean \pm SD) based on three independent cells. The corresponding error bars have been added to Figs. R3 and R4. These revisions have been incorporated into both the revised manuscript and the Supporting Information to enhance the reliability and reproducibility of the data.

Fig. R3 | Na-S battery performance at room temperature. a cycling performance at 1 A g⁻¹ using A/C-CoNiS/S and C-CoNiS/S. **b** Cycle stability at a high sulfur loading at 1 A g⁻¹. **c** rate performance of the two samples at current densities ranging from 0.5 to 5 A g⁻¹. **d** Long-term cycling stability of the A/C-CoNiS/S cathode at 3 A g⁻¹ over 1200 cycles.

In Fig. R3a, A/C-CoNiS/S shows a high capacity of 1422.3 mAh g⁻¹ after 250 cycles at 1 A g⁻¹ and a nearly 100 % coulombic efficiency (grey line). By contrast, C-CoNiS/S only achieves a discharge capacity of 694.4 mAh g⁻¹ with a capacity decay of 0.015 % per cycle. Even we increased the sulfur loading to 3 mg cm⁻², A/C-CoNiS/S can still give a capacity of 517.4 mAh g⁻¹ after 400 cycles at 1 A g⁻¹ and a nearly 100 % coulombic efficiency in Fig. R3b, surpassing most reported work (306~373 mAh g⁻¹). Furthermore, A/C-CoNiS/S in Fig. R3c delivers high specific capacities of 1493.7, 1435.0, 1389.2, 1318.4, and 1278.2 mAh g⁻¹ at 0.5,

1, 2, 3, 4, and 5 A g⁻¹, respectively. Even when the current density decreases to 0.5 A g⁻¹, the capacity can still be up to 1474.9 mAh g⁻¹, indicating excellent reversibility and cycling stability. Impressively, the A/C-CoNiS/S cathode in Fig. R3d delivers a high capacity of 1314.7 mAh g⁻¹ at 3 A g⁻¹ after 1200 cycles with capacity decay rate of 0.005 % per cycle and coulombic efficiency of 99.9%.

Fig. R4 | Na-S Battery Performance at low and high temperatures. a Rate performance of A/C-CoNiS/S from 0.2 to 5 A g⁻¹ at -20 °C. **b** Cycling performance for A/C-CoNiS/S and C-CoNiS/S at -20 °C. **c** Cycling performance of A/C-CoNiS/S and C-CoNiS/S at 50 °C.

Fig. R4a shows the rate performance of A/C-CoNiS/S at -20 °C. The A/C-CoNiS/S electrode delivers high specific capacities of 1036.7, 1000.0, 945.4, 907.4, 879.0, and 843.3 mAh g⁻¹ at current densities of 0.5, 1, 2, 3, 4, and 5 A g⁻¹, respectively. When the current density restores to 0.5 A g⁻¹, the discharge capacity can still be up to 1024.3 mAh g⁻¹ with almost no capacity loss. For comparison, the capacity of C-CoNiS/S decreases dramatically at 2 A g⁻¹, showing no future application in low-temperature batteries. A/C-CoNiS/S provides a high capacity of 957.4 mAh g⁻¹ after 400 cycles at 2 A g⁻¹ at -20 °C in Fig. R4b, and coulombic efficiency of nearly 100 %. By contrast, C-CoNiS/S delivers 521.6 mAh g⁻¹ after 140 cycles at 1 A g⁻¹. Fig. R4c shows A/C-CoNiS/S gives an initial specific capacity of 1424.0 mAh g⁻¹ at 2 A g⁻¹, which slightly decreases to 1402.6 mAh g⁻¹ after 700 cycles with an average capacity decay rate of 0.002 % per cycle at 50 °C. For comparison, C-CoNiS/S gives an initial discharge capacity of 825.9 mAh g⁻¹ with an 81.2 % retention rate after 120 cycles.

Related Revision:

(Manuscript, page 8, 10)

The sudden capacity loss of C-CoNiS/S might be caused by short circuit influenced by shuttle effect in the battery at high temperatures. The above battery data were obtained from three independent parallel tests to ensure reproducibility (Supplementary Fig. 20).

Impressively, the A/C-CoNiS/S cathode in Fig. 2g delivers a high capacity of 1320.8 mAh g⁻¹ at 3 A g⁻¹ after 1200 cycles with capacity decay rate of 0.012 % per cycle and coulombic efficiency of 99.9%. The above battery data were obtained from three independent parallel tests to ensure reproducibility (Supplementary Fig. 15).

(Supplementary information, page 17, 23)

Supplementary Fig. 15 | Na-S battery performance at room temperature. a cycling performance at 1 A g⁻¹ using A/C-CoNiS/S and C-CoNiS/S. **b** Cycle stability at a high sulfur loading at 1 A g⁻¹. **c** rate performance of the two samples at current densities ranging from 0.5 to 5 A g⁻¹. **d** Long-term cycling stability of the A/C-CoNiS/S cathode at 3 A g⁻¹ over 1200 cycles. Data from three parallel cells are presented, confirming the reproducibility of the battery performance.

Supplementary Fig. 20 | Na-S battery performance at low and high temperatures. a Rate performance of A/C-CoNiS/S from 0.2 to 5 A g⁻¹ at -20 °C. **b** Cycling performance for A/C-CoNiS/S and C-CoNiS/S at -20 °C. **c** Cycling performance of A/C-CoNiS/S and C-CoNiS/S at 50 °C. Data from three parallel cells are presented, confirming the reproducibility of the battery performance.

Comment 2-8: *The complete cyclic voltammetry curve (charge and discharge) in Figure 3a should be provided by the authors.*

Response 2-8: We thank the reviewer for the valuable suggestion. As the full CV curves at 25 °C have been shown in Fig. 4a in the previous version, we here supplement the complete CV curves at -20 °C and provide detailed explanations accordingly. Please check our revisions in the manuscript and SI.

Related Revision:

(Manuscript, page 8–9)

Fig. 3a shows the change of CV peaks between room temperature (25 °C) and low temperature (-20 °C). As for the selected oxidation peak of O₂ (Na₂S→Na₂S₄), it's clear that decreased temperature only makes A/C-CoNiS/S slightly shift by 0.09 V while makes C-CoNiS/S shift more positively by 0.20 V, indicating A/C-CoNiS provides a much easier conversion process for NaPSs at low temperatures. The complete CV curves at 25 °C and -20 °C were shown in Fig. 4a and Supplementary Fig. 16, respectively.

Supplementary Fig. 16 | CV curves of A/C-CoNiS/S and C-CoNiS/S at 0.2 mV s⁻¹ at -20 °C.

Four prominent cathodic peaks are observed at 1.91 V (R₀), 1.43 V (R₁), 0.89 V (R₂), and 0.48 V (R₃), along with a distinct peak at 0.61 V (R_M). These peaks are sequentially associated with the stepwise reduction of sulfur species: R₀ corresponds to the initial conversion of S₈ into long-chain polysulfides (Na₂S_x, 6 < x ≤ 8); R₁ to the transition from long-chain Na₂S_x to Na₂S₄; R₂ and R₃ to the further reduction of Na₂S₄ to Na₂S. The peak R_M is attributed to Na⁺ insertion into CoNiS²². During the anodic sweep, the peaks labeled O₁ to O₄ are assigned to the progressive reoxidation of Na₂S back to Na₂S_x. The peak O_M arises from the extraction of Na⁺. Furthermore, A/C-CoNiS/S exhibits a larger current response and smaller polarization, indicating its excellent performance at -20 °C.

Comment 2-9: *There is a lack of research summary on amorphous crystalline structures in the field of metal–sulfur batteries. It is recommended to add the latest research results on related materials.*

Response 2-9: We thank the reviewer for the suggestion. We have incorporated a summary of the latest research advances on crystalline-amorphous heterostructures in the field of metal–sulfur batteries, with a focus on their role in enhancing polysulfide adsorption and catalytic conversion kinetics. Specifically, we have discussed recent works on: Phosphide-based

amorphous-crystalline systems ($\text{Ni}_2\text{P}/\text{CeO}_x$, $\text{P-CoS}_2/\text{CNT}$) (*Energy Storage Mater.* **2024**, *70*, 103551, *Chem. Eng. J.* **2024**, *70*, 150696), Oxide-based heterostructures (c-a- MoO_3) (*Nano Energy* **2025**, *133*, 110508). These heterostructures enhance performance by synergistically optimizing intermediate adsorption and conversion: amorphous regions provide abundant defects and active sites, strengthening adsorption of soluble intermediates (e.g., NaPSs/LiPSs) and accelerating electron/ion transport; crystalline regions offer ordered lattices and conductive pathways while buffering volume changes to improve cycling stability; and interfaces enable charge redistribution and built-in fields that lower reaction barriers and enhance catalytic activity.

These additions strengthen the context of our study and highlight the novelty of our approach in designing highly efficient electrocatalysts for metal–sulfur batteries. The revised introduction now better situates our work within the current state-of-the-art.

Related Revision:

(Manuscript, page 3–4)

Interface engineering can be one feasible strategy to further promote the catalytic and adsorption performances towards polysulfides^{17,18}. Among different kinds of interfaces, the amorphous/crystalline one could be an ideal structure for Na–S batteries. It combines the advantages of both phases, including the high conductivity and structural stability of crystalline regions as well as the abundant active sites and strong adsorption capacity of amorphous regions¹⁷⁻²¹. For instance, Lee et al. constructed a nanoscale crystalline–amorphous MoO_3 heterostructure, which facilitated continuous adsorption and conversion of lithium polysulfides, leading to high sulfur utilization and exceptional cycling stability¹⁹. Moreover, Liu et al. demonstrated that a crystalline–amorphous $\text{Ni}_2\text{P}/\text{CeO}_x$ heterostructure with strong interfacial electronic interaction could accelerate Li_2S deposition/decomposition kinetics and improve long-term cycle life²⁰.

Comment 2-10: *There are many type/grammar errors and also images in this manuscript, for example, in line 275, red and green, and there is no green in the image. In Figure 4c, the A/C-CoNiS electrode has a higher electrochemical impedance, and there are errors in plotting and expression.*

Response 2-10: We appreciate the careful review. All relevant errors have been addressed. In addition, we have thoroughly reviewed the entire manuscript and corrected similar issues to ensure overall consistency and accuracy. Due to the large number of changes, we do not list each one here, but all revisions have been highlighted in the revised manuscript and SI.

Reviewer #3 (Remarks to the Author):

General Comments: *The manuscript presents a compelling advancement in Na–S battery catalysts. Strengthening the link between Sn doping, structural modulation, and performance, along with contextualizing claims relative to existing literature, would elevate its impact. Addressing the following issues will ensure broader accessibility. Some comments need to be addressed before publication:*

Response: Thank you very much for your valuable comments and suggestions, which greatly contributes to improving the quality of our manuscript. We have addressed your comments one by one as follows.

Comment 3-1: *Pore size and surface area not clearly stated in the text. Is the structure micro/meso/macroporous?*

Response 3-1: We appreciate the reviewer's insightful question. Nitrogen adsorption–desorption (BET) tests were performed to evaluate the materials' specific surface area and pore size distribution. The results show that A/C-CoNiS exhibits a higher specific surface area ($35.3 \text{ m}^2 \text{ g}^{-1}$) and a mesoporous structure with a dominant pore size of approximately 3.9 nm (Supplementary Fig. 9). This mesoporous architecture enhances electrolyte infiltration, accelerates ion transport, and provides abundant active sites, thereby contributing to the improved electrochemical performance. In response to the reviewer's comments, we have incorporated this discussion into the revised manuscript and SI.

Related Revision:

(Manuscript, page 6)

In addition, nitrogen adsorption–desorption analysis confirms that A/C-CoNiS possesses a larger surface area and a mesoporous structure (~3.9 nm), which facilitates electrolyte infiltration and ion diffusion (Supplementary Fig. 10).

(Supplementary information, page 12)

Supplementary Fig. 10 | BET characterization of A/C-CoNiS and C-CoNiS. a N₂ adsorption–desorption isotherms. **b** Pore size distribution calculated by the BJH model. Supplementary Fig. 10a shows that A/C-CoNiS exhibits a BET specific surface area of 35.3 m² g⁻¹, much higher than that of C-CoNiS (10.8 m² g⁻¹). The BJH pore size distribution in Supplementary Fig. 10b shows dominant mesopores centered at 3.9 nm for A/C-CoNiS and 4.3 nm for C-CoNiS, confirming a typical mesoporous structure. The two samples have similar pore size but A/C-CoNiS has more pores by comparing the pore volume. This mesoporous architecture facilitates electrolyte infiltration, accelerates ion transport, and provides abundant active sites, contributing to the improved electrochemical performance.

Comment 3-2: A Whatman glass fiber separator is used instead of a typical Celgard polymer separator. The former is much thicker and will reduce energy density. What is the cycling performance using Celgard separator?

Response 3-2: We appreciate the reviewer’s valuable suggestion. We performed cycling tests using a PE separator (Fig. R5). At a current density of 0.2 A g⁻¹, A/C-CoNiS/S delivered a discharge capacity of 1085.0 mAh g⁻¹ after 60 cycles, with an average decay rate of 0.29% per cycle. Additionally, the Coulombic efficiency of the battery is close to 100%. The results demonstrate that the battery can also operate stably using a typical Celgard polymer separator.

Fig. R5 | Cycling performance of the A/C-CoNiS/S using a PE separator.

Comment 3-3: *There are many reports on sulfur host materials and it is currently known that anode and electrolyte are more of a problem in these battery systems than new cathode host materials. And it is always a bit suspicious if the one and only solution comes up with a new quite exotic composite material.*

Response 3-3: We thank the reviewer for the thoughtful comment. We acknowledge that the anode and electrolyte are crucial to the battery, but we are convinced that research on the cathode catalyst is also important. We believe that our catalyst-oriented cathode design provides a valuable and complementary approach that contributes to the overall performance enhancement of practical room-temperature Na–S batteries.

The cathode of the Na–S battery involves redox reactions with 16 electrons. The slow reaction kinetics limit the rate performance of Na–S batteries, especially under low-temperature operating conditions. Introducing an efficient catalyst at the cathode can effectively promote the transformation of sulfur species, thereby improving its rate performance. Additionally, Na–S batteries also suffer from a severe shuttle effect of polysulfides, which is aggravated at high temperatures. By introducing rationally designed materials at the cathode, the anchoring of polysulfides can optimize the cycling stability of the battery.

As a result, our study focuses specifically on the cathode side, aiming to alleviate sluggish redox kinetics and suppress the shuttle effect through catalyst-oriented design. While this does not address all system-level issues, we believe it offers meaningful insights into improving cathode performance, which remains an essential component of overall battery efficiency. The excellent electrochemical performance across a wide temperature range demonstrated in the manuscript proves that our strategy is highly effective.

Comment 3-4: *It might be a good idea to quantify the adsorption capacity with UV Vis following the method reported by Hippauf. This would give a good comparison with other more polar carbon host materials as reported by Hao. Also a water isotherm would be helpful to check the hydrophilic character of the host.*

Response 3-4: We sincerely appreciate the reviewer's constructive suggestion. **To address this, we quantitatively evaluated the polysulfide adsorption capacity of the catalysts using UV-vis spectroscopy (Supplementary Fig. 21). As shown in Supplementary Table 3, the adsorption capacities at 25 °C and 50 °C are 0.080 and 0.078 g g⁻¹ for A/C-CoNiS, and 0.056 and 0.020 g g⁻¹ for C-CoNiS, respectively.** A/C-CoNiS consistently exhibits superior polysulfide adsorption, with the advantage being most pronounced at 50 °C. These results underscore the excellent temperature-adaptive adsorption capability of A/C-CoNiS.

We thank the reviewer for the suggestion to evaluate the hydrophilicity of the material using a water adsorption isotherm. **However, since our system employs an ether-based electrolyte rather than an aqueous medium, water adsorption cannot accurately reflect the actual electrode-electrolyte interactions. Therefore, we used DEGDME as the probe liquid for contact angle measurements.** The results reveal that the droplet spreads rapidly and reaches a contact angle near zero (Fig. R6), indicating excellent wettability. Furthermore, BET analysis (Supplementary Fig. 9) confirms that A/C-CoNiS possesses a mesoporous structure with a

dominant pore size of approximately 3.9 nm. This mesostructure facilitates electrolyte infiltration, ion transport, and polysulfide confinement through capillary and interfacial effects. The consistent conclusions from the contact angle measurements and BET analysis demonstrate the strong affinity between A/C-CoNiS and DEGDME.

Please check the corresponding revisions in the manuscript and SI.

Fig. R6 | Contact angle measurement between DEGDME and the A/C-CoNiS surface.

Related Revision:

(Manuscript, page 12)

Fig. 4h shows that A/C-CoNiS exhibits a markedly weaker UV-vis absorption signal than C-CoNiS and the Na₂S₆ solution at room temperature, revealing superior adsorption ability. Quantitative UV-vis adsorption tests (Supplementary Figs. 24 and 25, Table 4) confirm that A/C-CoNiS exhibits superior adsorption capacities especially at high temperatures, indicating an improved anti-shuttle effect.

(Supplementary information, page 28–30)

Supplementary Fig. 24 | A linear calibration curve was established based on Na_2S_6 solutions of known concentrations. a UV-vis absorption spectra of Na_2S_6 solutions with varying concentrations. **b** Corresponding calibration curve used for quantitative analysis. Based on the established calibration curve, the adsorption capacity of the catalyst for polysulfides can be quantitatively determined from the corresponding UV-vis absorption data.

Supplementary Fig. 25 | UV-vis spectra of Na_2S_6 solutions after immersion with A/C-CoNiS and C-CoNiS at 50 °C for 12 h. Inset: digital photographs of the solutions before and after adsorption. Compared to C-CoNiS and the Na_2S_6 solution, A/C-CoNiS exhibits weaker UV-vis signals and a more transparent solution (inset), indicating stronger NaPSs adsorption and improved anti-shuttle capability ⁴.

Supplementary Table 4.

Polysulfide adsorption capacity of A/C-CoNiS and C-CoNiS at different temperatures.

Sample	Temperature (°C)	Adsorption Capacity (mmol·g ⁻¹)	Adsorption Capacity (g _s ·g ⁻¹)
A/C-CoNiS	25	0.336	0.080
	50	0.328	0.078
C-CoNiS	25	0.235	0.056
	50	0.086	0.020

(Manuscript, page 18)

Polysulfide adsorption tests

2.5 mM Na₂S₆ solution was prepared by dissolving stoichiometric amounts of sulfur and anhydrous sodium sulfide (Na₂S, AR) in 10 mL of DEGDME. The mixture was heated to 80 °C and stirred continuously for 48h in an argon-filled glove box. For adsorption experiments, 10 mg of powder (A/C-CoNiS or C-CoNiS) was added to 2 mL Na₂S₆ solution. After standing for 12h, the color change of the solution was recorded with a digital camera, and the supernatant was analyzed by UV-vis spectroscopy (after 10-fold dilution). The powders were then dried and characterized by XPS to investigate chemical interactions between the catalysts and polysulfides. To quantify adsorption capacity, UV-vis spectra of Na₂S₆ solutions at varying concentrations (0.05, 0.10, 0.15, 0.20, and 0.25 mM) were measured.

Comment 3-5: *Of course a big flaw are the limited experimental details. No electrolyte/sulfur ratio given. That's a step backwards in reporting accurate data. With a whatman separator (thick) an excess of electrolyte is guaranteed! The huge capacity loss in the first cycle confirms this view (dissolution). And in a flooded cell PS diffusion takes longer. And the glass fibres also adsorb the PS, and affect crystallization.*

It would be good to test the material also with a PE separator and compare it to a Ketjen Black based reference cathode. And check performance also under lean conditions with E/S ratios varying between 5 and 10.

Response 3-5: We sincerely thank the reviewers for their valuable feedback. In our experiments, the electrolyte-to-sulfur (E/S) ratio was $180 \mu\text{L mg}^{-1}$, and additional experimental details have been included in the revised manuscript to enhance the clarity and reproducibility of the study. To address the reviewer's concerns, we conducted a series of control experiments:

- Replacing the Whatman glass fiber separator with a PE separator (Fig. R5), A/C-CoNiS/S delivered a discharge capacity of $1085.0 \text{ mAh g}^{-1}$ after 60 cycles at 0.2 A g^{-1} , corresponding to an average decay rate of 0.29% per cycle.
- At 1 A g^{-1} , KB/S maintained a discharge capacity of 339.4 mAh g^{-1} after 250 cycles, after which the capacity declined sharply (Supplementary Fig. 12c).
- At an E/S ratio of $10 \mu\text{L mg}^{-1}$ (Supplementary Fig. 13), A/C-CoNiS/S retained a discharge capacity of $1146.9 \text{ mAh g}^{-1}$ after 50 cycles at 0.2 A g^{-1} .

According to the reviewers' comments, we have added a discussion of relevant discussion in the revised manuscript and SI.

Related Revision:

(Manuscript, page 7)

Supplementary Fig. 12a shows that the sodium-ion storage performance of A/C-CoNiS contributes minimally to the overall capacity, indirectly indicating that the main contribution comes from the redox process of sulfur. The MXene/S and Ketjen Black/S cathodes deliver capacities far below A/C-CoNiS/S, highlighting the catalytic advantage of A/C-CoNiS (Supplementary Fig. 12b and 12c).

(Supplementary information, page 14)

Supplementary Fig. 12 | Cycling stability of the A/C-CoNiS, MXene/S, and KB/S at 1 A g⁻¹. In sodium-ion batteries, A/C-CoNiS delivers 215.8 mAh g⁻¹ after 250 cycles at 1 A g⁻¹ (Supplementary Fig. 12a). Considering its 30 wt% content in the A/C-CoNiS/S composite, A/C-CoNiS contributes only 6.4% of the total capacity, confirming that sulfur is the primary capacity source. In Na-S batteries, MXene/S and KB/S electrodes deliver discharge capacities of 365.1 and 339.4 mAh g⁻¹ respectively after 250 cycles at 1 A g⁻¹ (Supplementary Figs. 12b, c), highlighting the superior catalytic performance of A/C-CoNiS.

(Manuscript, page 7)

Notably, the A/C-CoNiS/S cathode maintains a discharge capacity of 1146.9 mAh g⁻¹ after 50 cycles at 0.2 A g⁻¹ with an E/S ratio of about 10 μL mg⁻¹ (Supplementary Fig. 13).

(Supplementary information, page 15)

Supplementary Fig. 13 | Cycling performance of A/C-CoNiS/S at E/S ratio of 10 $\mu\text{L mg}^{-1}$.

Comment 3-6: *Why is cobalt used rather than another metal; is the choice of cobalt important?*

I don't think the choice of cobalt versus other transition metals is discussed in the paper and the computational authors could test other metal ions!

Response 3-6: Thank you very much for your valuable comments. Cobalt was chosen due to its unique electronic configuration and balanced d-orbital occupancy, which favorably regulate polysulfide adsorption and catalytic conversion during sulfur redox reactions. Previous density functional theory studies have shown that the occupation of e_g and t_{2g} orbitals in metal clusters plays a key role in polysulfide interaction (Fig. R7). The e_g/t_{2g} ratio effectively describes binding strength and catalytic activity. Electrons in e_g orbitals influence the σ_1^* and σ_3^* antibonding states related to S–S bond cleavage, where excessive occupancy may hinder bond breaking. t_{2g} electrons affect the σ_3^* state, determining adsorption strength; high t_{2g} occupancy may lead to overly strong binding and hinder desorption. Among the tested metals (Fe, Co, Ni, Cu, Zn), cobalt shows a moderate e_g/t_{2g} ratio of 0.312, between Fe (0.301) and Ni (0.328). This balance helps avoid both over-retention (as in Fe) and weak adsorption (as in Zn), ensuring efficient polysulfide conversion and contributing to the improved SRR kinetics and cycling stability observed. (*Nat. Nanotechnol.* **2024**, *19*, 792-799)

Electrochemical measurements (Fig. R8) reveal that Co exhibits a higher diffusion-limited current density than Fe, but lower than Cu and Zn, indicating a moderate yet stable catalytic activity. This balance allows Co to sustain efficient polysulfide conversion at high current densities while avoiding the side reactions typically associated with overactive catalysts like Cu and Zn or the sluggish kinetics observed with Fe due to insufficient activity. (*Nat. Nanotechnol.* **2024**, *19*, 792-799)

Based on the unique advantages of Co mentioned above, we choose Co here. According to the reviewers' comments, we have added a discussion of relevant discussion in the revised manuscript.

Fig. R7 | Correlation between polysulfide concentration and molecular orbitals of catalysts. **a**, Schematic for the degeneracy of molecular orbitals between polysulfides and catalysts. **b**, Projected DOS plots for different orbitals for Fe, Co, Ni, Cu and Zn catalysts (top to bottom). The x axis values are the energy levels subtracting the Fermi levels for each catalyst.

c, Synchrotron-based L_3 -edge NEXAFS plots for the different catalysts. **d**, Theoretical and experimental confirmation of the SRR kinetic trend with DOS-based and NEXAFS-based e_g/t_{2g} ratios for the different catalysts. The error bars indicate ± 2 nm wavelength differences used to calculate the Li_2S_4 concentrations.

Fig. R8 | **a**, LSV curves for the different catalysts. J_{geo} is the current density based on the geometric area of electrode. **b**, Relationship between J_d and Li_2S_4 concentration. The error bars indicate the ± 2 nm wavelength difference used to calculate the polysulfide concentration. The dotted line is the linear fitting for the data points, and R^2 is the coefficient of determination for linear fitting.

Related Revision:

(Manuscript, page 3)

Transition metal sulfides have been proven to be efficient catalysts for metal-sulfur batteries¹², which might also be favorable for low-temperature batteries. Cobalt's unique electronic structure enables a balance between polysulfide adsorption and conversion, ensuring stable catalytic performance across wide temperatures¹³⁻¹⁵.

Reviewer #4 (Remarks to the Author):

General Comments: *This manuscript presents a comprehensive study aimed at mitigating the polysulfide shuttle effect in Na–S batteries through the design and application of a Sn-doped CoNiS catalyst. By integrating both experimental investigations and density functional theory (DFT) calculations, the authors have made commendable efforts to elucidate the role of the electrocatalyst in enhancing the electrochemical performance of Na–S batteries. While the study offers promising insights and contributes to the development of effective shuttle suppression strategies, several critical aspects require further clarification and elaboration. In particular, addressing certain mechanistic details and expanding the discussion of computational findings will significantly strengthen the depth and impact of the work. Specific comments and suggestions are provided below to help improve the overall quality and completeness of the manuscript.*

Response: Thank you very much for your valuable comments and suggestions, which have greatly contributed to improving the quality of our manuscript. We have addressed your comments one by one as follows.

Comment 4-1: *In this study involving Na_2S_x species ($x = 8, 6, 4, 2,$ and 1), the authors mention the use of a $1 \times 1 \times 1$ Monkhorst-Pack Gamma-centered k -point mesh and a plane-wave energy cutoff of 400 eV for all calculations. Could the authors please clarify the rationale behind selecting the Gamma point for k -point sampling and the specific cutoff energy value? Additionally, did the authors perform any convergence tests with respect to the k -point mesh and energy cutoff to ensure the reliability and accuracy of the computed results?*

Response 4-1: We sincerely appreciate the reviewer's valuable suggestion. In the initial calculations, such a setup is commonly employed for large supercells and surface slab systems, as it provides a reasonable balance between computational cost and accuracy. To further ensure

the reliability of our results, we have conducted additional convergence tests by increasing the k-point mesh to $1 \times 2 \times 1$ and the plane-wave cutoff energy to 500 eV. According to the reviewers' comments, the relevant data and corresponding discussion have been incorporated into the revised manuscript and SI.

Related Revision:

(Manuscript, page 18)

Theoretical calculations

All first-principles calculations were conducted using the Vienna Ab initio Simulation Package (VASP). The electron exchange-correlation energy was described using the Perdew–Burke–Ernzerhof (PBE) exchange-correlation functional within the generalized gradient approximation (GGA) framework. To accurately describe the strong electron correlation effects in transition metals, the DFT+U method was employed with Hubbard U parameters of 3.32 eV for Co, 6.20 eV for Ni, and 5.80 eV for Ti^{52,53}. The empirical dispersions of Grimme (DFT-D3) were applied to account for the long-range van der Waals interactions⁴⁹. To align with experimental characterizations, the (311) surface was cleaved from the $(\text{Co}_{0.5}\text{Ni}_{0.5})_9\text{S}_8$ structure. For calculations involving Na_2S_x ($x = 8, 6, 4, 2,$ and 1), a $1 \times 2 \times 1$ Monkhorst-Pack Gamma-centered k-point mesh was employed. The cutoff energy was set to 500 eV for all calculations, with convergence criteria of $0.01 \text{ eV}\text{\AA}^{-1}$ for forces and 10^{-5} eV for energy. The adsorption energy (E_{ads}) was calculated using the following equation, $E_{\text{ads}} = E_{\text{adsorb/surf}} - E_{\text{surf}} - E_{\text{adsorb}}$, where $E_{\text{adsorb/surf}}$ represents the total energy of the adsorbed system, E_{surf} is the energy of the optimized clean substrate, and E_{adsorb} denotes the energy of the adsorbate in vacuum.

Comment 4-2: *Considering the presence of transition metal atoms with localized d-electrons in your system, did the authors consider employing the DFT+U approach to better account for on-site Coulomb interactions? If not, could you please elaborate on the justification for*

omitting the Hubbard U correction, and whether any test calculations were conducted to assess its potential impact on the electronic structure and total energy?

Response 4-2: We would like to thank the reviewer for the valuable comments. In the DFT calculations, we have revised the Gibbs free energy results by applying the Hubbard U correction. Specifically, test calculations were conducted by applying Hubbard U corrections to Co (U = 3.32 eV, *Nat. Catal.* **2024**, 7, 1213) and Ni (U = 6.20 eV, *Nat. Catal.* **2024**, 7, 1213). We have updated the calculation results in the manuscript and added the relevant descriptions in the theoretical simulation section.

Related Revision:

(Manuscript, page 18)

To accurately describe the strong electron correlation effects in transition metals, the DFT+U method was employed with Hubbard U parameters of 3.32 eV for Co, 6.20 eV for Ni, and 5.80 eV for Ti^{52,53}. The empirical dispersions of Grimme (DFT-D3) were applied to account for the long-range van der Waals interactions⁴⁹.

(Manuscript, page 29)

Fig. 5 | Boosted catalytic kinetics and adsorption ability at the amorphous/crystalline interface revealed by DFT calculations. a Atomic structure model of A/C-CoNiS. **b** Model of the counterpart C-CoNiS with crystalline structure. Step chart of free energy with active site from **c** amorphous region of A/C-CoNiS (Amor-A/C-CoNiS) and **d** crystalline region of A/C-CoNiS (Cryst-A/C-CoNiS). Both **c** and **d** use C-CoNiS for comparison. Energy profiles for Na₂S decomposition on **e** Amor-A/C-CoNiS and **f** Cryst-A/C-CoNiS. Both **e** and **f** use C-CoNiS for comparison. **g** Atomic structure configurations of NaPSs and S₈ adsorbed on the surface of A/C-CoNiS. **h** Adsorption energies of S₈ and NaPSs on A/C-CoNiS and C-CoNiS. **i** Differential charge density distributions of S₈ adsorbed on A/C-CoNiS and C-CoNiS surface with isosurface value of 0.005 e Å⁻³.

Finally, we analyzed the influence of amorphous/crystalline interface in A/C-CoNiS on catalytic reaction kinetics by DFT calculations. As shown in Fig. 5a, the atomic structure covers two parts, the amorphous structure (marked with Amor-A/C-CoNiS) generated by Sn doping, and the side parts with crystalline structure (Cryst-A/C-CoNiS). Fig. 5b shows the model of C-CoNiS with only crystalline structure for comparison. Because there are two parts in Fig. 5a, we had to select active sites from both amorphous and crystalline regions. Specifically, we screened multiple possible sites and identified the one with the lowest S_8 adsorption energy as the optimal active site (Supplementary Figs. 27–29). As shown in Figs. 5c and 5d, the rate-determining step (RDS) from Na_2S_2 to Na_2S shows a free energy value of 0.36 eV for C-CoNiS, which decreases to 0.25 eV for Amor-A/C-CoNiS and 0.27 eV for Cryst-A/C-CoNiS. That is, not only the amorphous region (Fig. 5c) but also the nearby crystalline region (Fig. 5d) in A/C-CoNiS show decreased energy barrier for NaPSs conversion. As shown in Figs. 5e and 5f, the climbing-image nudged elastic band (CI-NEB) calculations reveal that the decomposition barrier of Na_2S is significantly reduced from 0.77 (C-CoNiS) to 0.58 eV (Amor-A/C-CoNiS) and 0.68 eV (Cryst-A/C-CoNiS). The structural evolution of Na_2S decomposition is provided in Supplementary Figs. 30. Notably, both the amorphous regions and the neighboring crystalline regions of A/C-CoNiS facilitate Na_2S decomposition by lowering the energy barrier⁵⁰. These studies verify that the amorphous/crystalline interface generated by Sn doping optimize both local electronic microenvironment of metal sites in both amorphous and crystalline regions.

Comment 4-3: *In the DFT investigation of the A/C-CoNiS interface and its role in catalytic reaction kinetics, how did the authors determine the specific active site(s) for Na–S adsorption, particularly considering the structural heterogeneity and potentially numerous adsorption*

sites present in the amorphous region? Given that amorphous structures can host a variety of chemically distinct sites, was any systematic approach like energy screening was employed to identify the most representative or catalytically relevant sites?

Response 4-3: We sincerely appreciate the reviewer's valuable comments. A systematic screening of candidate sites on the C-CoNiS and A/C-CoNiS was conducted to precisely determine the possible adsorption sites relevant to catalysis. To rationally reduce complexity, we first grouped atoms into equivalent categories by comparing their charge values, where atoms with nearly identical charge numbers were considered to belong to the same chemical environment. Within each category, representative atoms were selected, and their adsorption energies toward S₈ were calculated (adsorption energies and adsorption configurations are shown in Supplementary Figs. 25–27). Among these, the site with the lowest adsorption energy was identified as the most catalytically relevant active site. We have added the relevant discussion in the revised manuscript and SI.

Related Revision:

(Manuscript, page 13)

Specifically, we screened multiple possible sites and identified the one with the lowest S₈ adsorption energy as the optimal active site (Supplementary Figs. 27–29).

(Supplementary information, page 32–34)

Supplementary Fig. 27 | Calculated adsorption free energy of S₈ on A/C-CoNiS at different adsorption sites.

Supplementary Fig. 28 | Calculated adsorption free energy of S₈ on C-CoNiS at different adsorption sites.

Supplementary Fig. 29 | Gibbs free energy profiles of S_8 adsorption at different sites on a C-CoNiS and **b** A/C-CoNiS. **Site Screening Methodology:** The S_8 molecule was found to preferentially adsorb on transition-metal (Co, Ni) sites rather than sulfur sites. Accordingly, all exposed transition-metal atoms were evaluated, and the site with the strongest binding energy was identified as the active site for further investigation. The corresponding adsorption configurations and energies were provided in Supplementary Figs. 25–27.

Comment 4-4: (a) The results indicate that the amorphous region of the A/C-CoNiS system exhibits the highest binding energies for Na-S adsorption compared to the crystalline counterpart. Could the authors provide further insights into the underlying factors responsible for these enhanced adsorption energies at the amorphous sites? A detailed discussion on the bonding nature—such as charge transfer characteristics, local coordination environments, or electronic structure analysis—would greatly aid in understanding the origin of the stronger interaction.

(b) Was any assessment of the system's electronic conductivity performed to evaluate its influence on the overall catalytic performance? Given that electronic transport plays a crucial role in redox kinetics, such analysis would be of significant value.

(c) In addition, the manuscript briefly mentions Sn doping in the catalyst. Could the authors elaborate on the specific role of Sn in modifying the catalyst's properties? In particular, does Sn incorporation induce any notable changes in the electronic structure or charge redistribution that could influence Na-S adsorption or reaction kinetics?

Response 4-4: We appreciate the reviewer's insightful comment.

(a) We examined the charge density differences during Na₂S₆ adsorption. As illustrated in Fig. 5i, a more substantial charge transfer occurs on A/C-CoNiS, which is indicative of stronger electronic interactions. The result suggests that the unique local coordination and structural disorder of the amorphous regions strengthen polysulfide binding, thereby explaining the higher adsorption energies and more effective shuttle suppression observed for A/C-CoNiS (*Adv. Mater.* **2024**, *36*, 2411725).

(b) Four-probe measurements were conducted to determine the intrinsic conductivities of the catalysts, which were found to be 61.0 S m⁻¹ for A/C-CoNiS and 50.7 S m⁻¹ for C-CoNiS. The higher conductivity of A/C-CoNiS can be attributed to the presence of a small amount of MXene, which possesses high intrinsic conductivity (*Nano-Micro Lett.* **2025**, *17*, 93), and a greater concentration of sulfur vacancies (*Angew. Chem. Int. Ed.* **2024**, e202404816) in A/C-CoNiS, both of which facilitate electron transport.

(c) We performed differential charge density analysis for Sn-doped A/C-CoNiS. As shown in Fig. R9, introduction of Sn leads to electron gain by neighboring Co and Ni atoms, consistent with XPS observations (decreased Co/Ni valence). Sn doping induces pronounced charge delocalization, especially in the amorphous regions, creating highly active sites that enhance adsorption effect and improve redox kinetics (*Adv. Mater.* **2025**, 2506839).

Fig. R9 | Differential charge density map of Sn-doped A/C-CoNiS. Yellow regions indicate electron accumulation, while cyan regions indicate electron depletion (Isosurface value is 0.005 e A^{-3}).

Comment 4-5: *The reversibility of the Na–S battery is a critical parameter influencing long-term performance. In this context, could the authors provide insights into the sodium diffusion mechanism during the charging process? Specifically, how does the Na–S bond cleavage proceed on the catalyst surface, and what are the associated kinetic barriers? It would be helpful if the authors could support their discussion with a kinetic analysis—such as nudged elastic band (NEB) calculations—to quantify the energy barriers involved in Na diffusion and bond dissociation. Was such an analysis performed or considered?*

Response 4-5: We sincerely appreciate the reviewer’s constructive suggestion. To elucidate the energy barrier associated with Na_2S decomposition, we performed climbing-image nudged elastic band (CI-NEB) calculations. The decomposition energy barriers of Na_2S on Amor-A/C-CoNiS, Cryst-A/C-CoNiS, and C-CoNiS are calculated to be 0.58 eV, 0.68 eV, and 0.77 eV, respectively (Figs. 5e and 5f). These results suggest that A/C-CoNiS, with its abundant catalytically active sites, can more effectively facilitate the cleavage of Na–S bonds, thereby accelerating the oxidative decomposition of Na_2S . The calculation results further confirm the enhanced electrocatalytic activity of A/C-CoNiS. According to the reviewers’ comments, we added NEB energy profiles and relevant discussion in the revised manuscript.

Related Revision:

(Manuscript, page 13)

As shown in Figs. 5e and 5f, the climbing-image nudged elastic band (CI-NEB) calculations reveal that the decomposition barrier of Na₂S is significantly reduced from 0.77 eV (C-CoNiS) to 0.58 eV (Amor-A/C-CoNiS) and 0.68 eV (Cryst-A/C-CoNiS). The structural evolution of Na₂S decomposition is provided in Supplementary Fig. 30. Notably, both the amorphous regions and the neighboring crystalline regions of A/C-CoNiS facilitate Na₂S decomposition by lowering the energy barrier⁵⁰.

(Supplementary information, page 35)

Supplementary Fig. 30 | CI-NEB calculation of the dissociation pathway of Na₂S on the a Amor-A/C-CoNiS, b Cryst-A/C-CoNiS, and c C-CoNiS surface.

Comment 4-6: *MXene has been previously reported to strongly adsorb Na–S species, which may result in structural integrity challenges in Na–S battery systems. Given its role as a substrate in your catalyst design, did the authors consider or evaluate the potential charge transfer between the MXene and the active catalytic components? Such electronic interactions could significantly influence the adsorption strength, electronic structure, and catalytic activity. An analysis of charge redistribution (e.g., Bader charge analysis or density of states) would provide valuable insights into the impact of the MXene support on the overall catalytic behavior.*

Response 4-6: We appreciate the reviewer’s insightful comments regarding the interfacial effects of MXene. It is well established that MXene can modulate interfacial electronic

structures at the nanoscale. Our previous work shows that 5 nm NiCoP nanoparticles on MXene nanosheets exhibit strong interfacial coupling, optimizing surface adsorption and accelerating intermediate conversion (*ACS Nano* **2022**, *16*, 11049–11058). However, the catalyst particles in our work are approximately 800 nm in diameter and supported on MXene sheets, resulting in limited interfacial contact. Consequently, charge transfer between catalyst and substrate is significantly weakened and not the dominant factor. In addition, the main role of MXene is to provide steric hindrance to induce the formation of multipods catalyst morphology. The key innovation of this work lies in the precise engineering of the catalyst's intrinsic structure—specifically the synergy between amorphous/crystalline interfaces and hierarchical architecture construction—to enhance catalytic activity, rather than the modulating effect of MXene on the catalytic performance of sulfides. Based on the above considerations, to better align with the structural characteristics of the material, we do not discuss the electronic interactions between MXene and the catalyst.

Response Letter for Manuscript “Amorphous-crystalline interwoven multipods with high Co/Ni activity for sodium-sulfur battery operated from –20 to 50 °C”

Thank you for reading this letter. We are grateful to the editor for providing us with the opportunity to further revise our manuscript, and we extend our heartfelt thanks to all reviewers for their constructive comments and positive evaluations. To address the points raised by Reviewer #1, we have added comparative SEM images and additional electrochemical measurements with varying catalyst-to-cathode ratios. We have carefully considered all feedback and revised the manuscript and Supplementary Information accordingly.

Reviewer #1 (Remarks to the Author):

General Comments: *The revised manuscript has well addressed my questions. I recommend it for publication pending the following supplementary information:*

Response: Thank you for your positive evaluation and constructive comments. We respond to your comments one by one as follows. Thanks again.

Comment 1-1: *Does Sn doping induce morphological changes in CoNiS? Comparative SEM analysis before and after doping is required.*

Response 1-1: We thank the reviewer for this valuable suggestion. Sn doping indeed induces noticeable morphological changes in CoNiS. As shown in Supplementary Fig. 8, the pristine CoNiS consists of interconnected spherical particles with an average diameter of approximately 1 μm. After Sn incorporation, the material evolves into a unique multipod structure, accompanied by interwoven crystalline and amorphous domains (Fig. 1b and Supplementary Fig. 3a). These results confirm that Sn doping plays a critical role in tuning the morphology and structural disorder of CoNiS. In response to the reviewer’s comment, we have incorporated

related SEM characterization into the revised manuscript and Supplementary Information, together with the relevant discussion, to provide a more complete and direct morphological comparison.

Related Revision:

(Manuscript, page 5)

It's interesting to find that Sn introducing not only induces the formation of unique multipod structure, but also leads to the interwoven crystalline and amorphous structure.

(Supplementary information, page 4)

Supplementary Fig. 3 | Characterizations of A/C-CoNiS. a SEM image and b XRD pattern

of A/C-CoNiS. Supplementary Fig. 3a exhibits a distinctive multipod morphology with radially extended branches.

In Supplementary Fig. 3b, the diffraction peaks match well with cubic Co₉S₈ with planes of (111), (311), (222), (331), (511), and (440) (JCPDS No. 19-0364). The synthesized (CoNi)₉S₈ by adding nickel elements has the same atomic structure as that of Co₉S₈. The low degree of crystallinity indirectly confirms the amorphous-crystalline interwoven structure.

(Supplementary information, page 8)

Supplementary Fig. 8 | Characterizations of C-CoNiS. **a** SEM image and **b** XRD pattern of C-CoNiS. Supplementary Fig. 8a exhibits some linked spheres with an average diameter of 1 μm . In Supplementary Fig. 8b, the diffraction peaks match well with cubic Co_9S_8 structure with planes of (111), (311), (222), (331), (511), and (440) (JCPDS No. 19-0364). The sharp peaks show high degree of crystallinity.

Comment 1-2: *What is the catalyst-to-cathode ratio? Is this the optimal ratio? Additional electrochemical performance data with varying catalyst ratios should be provided.*

Response 1-2: We thank the reviewer for this insightful comment. **The catalyst-to-cathode ratio used is 3:7**, derived from the sulfur content (70 wt%) of the A/C-CoNiS/S composite employed during cathode fabrication (see “Preparation of sulfur cathodes” section). To address the reviewer’s concern, we added the cycling performance at 1 A g^{-1} for cathodes with sulfur contents of 60, 80, and 90 wt%, corresponding to catalyst-to-cathode mass ratios of 4:6, 2:8, and 1:9, respectively (Supplementary Fig. 11). The results reveal no appreciable capacity difference at sulfur loadings of 60 % and 70 %. However, once the sulfur content reaches 80 % or 90 %, a discernible capacity fade is observed. While preserving the specific capacity, increasing the sulfur loading as much as possible is beneficial for enhancing the overall capacity and energy density of the cell. **Therefore, we selected a sulfur loading of 70 wt%, which corresponds to a catalyst-to-cathode mass ratio of 3:7.** The related discussion has been added to the revised manuscript and Supplementary Information.

Related Revision:

(Manuscript, page 6)

Supplementary Fig. 11 shows that A/C-CoNiS/S achieves its optimal electrochemical performance at a sulfur loading of 70 wt%, while the maximum sulfur loading of C-CoNiS/S is 65 wt%.

(Supplementary information, page 13)

Supplementary Fig. 11 | a TGA profiles of A/C-CoNiS/S. **b** Cycling performance of A/C-CoNiS/S at sulfur loadings of 60 wt%, 70 wt%, 80 wt%, and 90 wt%. **c** TGA profile of C-CoNiS/S. TGA tests were conducted over the temperature range of 30 to 400 °C with a heating rate of 10 °C min⁻¹. No significant mass loss was observed before 55 °C. The rapid weight loss observed between 150 and 350 °C is attributed to the volatilization of sulfur. TGA measurements on A/C-CoNiS/S prepared at 9:1, 8:2, 7:3, and 6:4 mass ratios yield sulfur loadings of 90, 80, 70, and 60 wt%, respectively. Electrochemical cycling at 1 A g⁻¹ (Supplementary Fig. 11b) shows that A/C-CoNiS/S with sulfur loadings of 60, 70, 80, and 90 wt% delivers capacities of 1419.9, 1417.0, 1323.6, and 1269.1 mAh g⁻¹ after 50 cycles. The results clearly show no appreciable capacity difference at sulfur loadings of 60 % and 70 %. However, once the sulfur content reaches 80 % or 90 %, a discernible capacity fade is observed. Therefore, we selected a sulfur loading of 70 % for A/C-CoNiS/S. Furthermore, TGA measurement on C-CoNiS/S prepared at a 7:3 mass ratio show that the sulfur loading is only 65 wt% due to its limited adsorption capability (Supplementary Fig. 11c).